# PLANNING WITH GENERATIVE COGNITIVE MAPS

## ABSTRACT

Planning relies on cognitive maps – models that encode world structure given cognitive resource constraints. The problem of learning functional cognitive maps is shared by humans, animals and machines. However, we still lack a clear understanding of how people represent maps for planning, particularly when the goal is to support cost-efficient plans. We take inspiration from theory of compositional mental representations in cognitive science to propose GenPlan: a cognitively-grounded computational framework that models redundant structure in maps and saves planning cost through policy reuse. Our framework integrates (1) a Generative Map Module that infers generative compositional structure and (2) a Structure-Based Planner that exploits structural redundancies to reduce planning costs. We show that our framework closely aligns with human behavior, suggesting that people approximate planning by piecewise policies conditioned on world structure. We also show that our approach reduces the computational cost of planning while producing good-enough plans, and contribute a proof-of-concept implementation demonstrating how to build these principles into a working system.

## 1 INTRODUCTION

People are highly proficient in solving real-world planning problems. For example, we can navigate cities without precisely knowing every link in the street network (Fig. 1a.)(Bongiorno et al., 2021) and accomplish complex construction projects (Fig. 1b.) with many actions and sub-goals (Mugan et al., 2024). Solving these problems optimally is theoretically intractable (Kaelbling et al., 1998), and therefore approximate algorithms for planning in natural domains remain an active area of research in AI (Silver & Veness, 2010), robotics (Curtis et al., 2025), and cognitive science (Kryven et al., 2024; van Opheusden et al., 2023). Here, we seek to uncover cognitive computations that enable humans to plan efficiently in natural domains. To do this, we focus on the key intuition that the human world is structured (Fig.1c,d) and propose that people reason about redundancies in this structure to efficiently encode cognitive maps, and reduce planning costs. We formalize this hypothesis in GenPlan, a computational model that gives an algorithmic account of how structure-based planning can be implemented in practice.

Formally, a planning problem constitutes a search within a decision tree of possible states and actions (Kuperwajs et al., 2025; Russell & Norvig, 2016). This tree can be encoded as a learned neural policy (Liu et al., 2020), an explicit tree structure (Russell & Norvig, 2016; Silver & Veness, 2010), or a model describing states and actions in a symbolic form (Tang et al., 2024). The size of the underlying state-space determines the computational cost of the problem, or how difficult it should be. Since optimal planning beyond non-trivial state-spaces is intractable, approximate planning frameworks have focused on building partial state-spaces (Silver & Veness, 2010), learning generalizable policies (Curtis et al., 2022; Singh et al., 2012), and grouping actions frequently performed together into options (Sutton et al., 1999). However, the difficulty predicted by these approximate planning algorithms rarely aligns with human experience, as people often solve formally complex real-world problems with relative ease.

We take inspiration from the theory of *compositional concepts* in cognitive science, which states that humans learn complex concepts by combining simpler ones (Fodor, 1975; Lake & Piantadosi, 2020; Pitt et al., 2021), and adapt the principle of compositionality to model human cognitive maps and plans as generative structures. Compositionality has been successful in explaining concept representation in visual (Lake & Piantadosi, 2020; Tian et al., 2020), auditory (Verhoef et al., 2014;

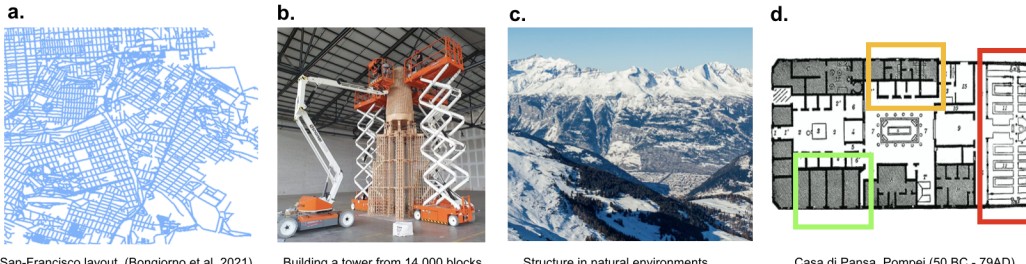

Figure 1: Structured human environments: city street networks, construction projects, natural landscapes, and an interior floor-plan with repeating structural elements highlighted. People learn mental world-models that exploit this structure to make resource-efficient plans.

Rohrmeier, 2020; Hofer et al., 2021) and spatial domains (Sharma et al., 2022; McCarthy et al., 2021). Further, compositional reasoning is culturally universal (Pitt et al., 2021), suggesting that it may be an evolved adaptation to natural structure people encounter in daily life (Johnston et al., 2022). Neural and behavioral studies provide ample evidence that cognitive maps are represented using similar compositional generative structures. Neural evidence from human studies includes mirror-invariant encoding of natural scenes (Dilks et al., 2011) and reuse of neural reference frames across similar environments (Marchette et al., 2014). Behavioral evidence includes hierarchical spatial representations (Kosslyn et al., 1974; Stevens & Coupe, 1978; Hirtle & Jonides, 1985) reflected in first planning routes between, and then within semantic regions (Bailenson et al., 2000; Newcombe et al., 1999; Wiener & Mallot, 2003; Wang & Brockmole, 2003; Balaguer et al., 2016; Tomov et al., 2020), and ability to predict unseen environment layout in structured environments (Sharma et al., 2022).

In formal terms, combinatorial concept representations can be modeled by *mental programs* – symbolic instructions specifying how to produce new instances of a given concept class (Lake et al., 2015; Lake & Piantadosi, 2020). Computational accounts of concept learning as *program induction* (inferring a program from a given a set of examples) provide powerful explanations of human learning efficiency – only a few examples can suffice to deduce an underlying program, in contrast to vast amounts of data required by purely neural models (Tenenbaum et al., 2011; Lake et al., 2015). Building on this research, we model cognitive maps as generative programs that capture structures such as symmetries and repeated parts, and propose an algorithmic framework that models cost-efficient planning in such maps by reusing local policy conditioned on structure, instead of solving a global optimization problem.

In this work we adopt a scientific and an engineering goal: (1) to understand computational cognitive principles by which humans plan in structured spatial domains, and (2) to engineer a cost-efficient computational framework that formalizes human-like planning in structured environments. We contribute:

- Generative Map Module (GMM), which discovers programmatic map representations using tractable inference;
- Structure-Based Planner (SBP) that implements hierarchical planning both within and between the structural units
- Empirical validation of our framework on human behavior, showing that human planning is consistent with generative cognitive maps and policy reuse.

The GMM models observations of the environment by inferring a small distribution over programmatic maps. To do this, we use a Large Language Model (LLM) as an embedding of human priors learned through training on human data. The SBP extends a Partially Observable Markov Decision Process (POMDP) to use the GMM representation. It constructs end-to-end policies for within-unit planning and between-unit transitions using adaptations of a Partially Observable Monte Carlo Planner (POMCP). In the next section, we introduce the experimental environment, followed by a detailed description of computational models. In Section 3, we compare our models' predictions with human empirical results.

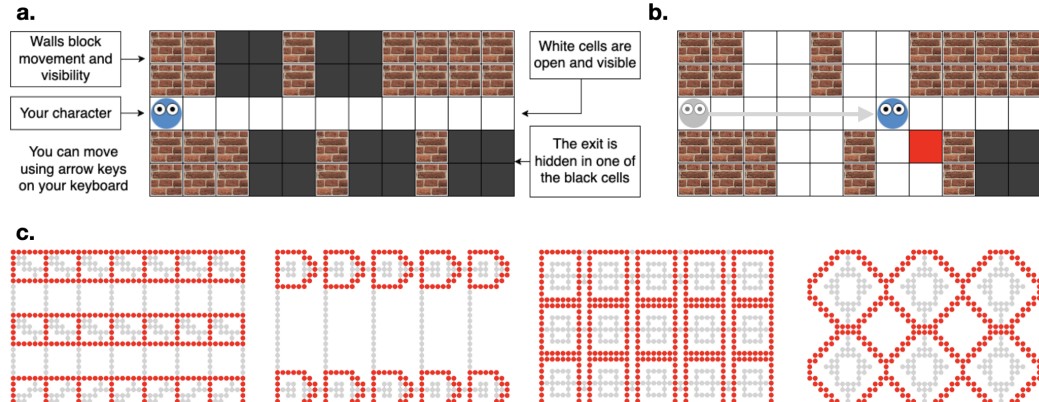

Figure 2: The Maze Search Task with structured layouts. (a.) Task Setup illustrated in a simple example. Participants can use keyboard keys to navigate over any non-wall cells. The exit is initially hidden in one of the black (unobserved) cells. (b) The exit is shown as a red tile when it comes into view. (c) A subset of structures environment layouts used to evaluate GenPlan and compare its performance to that of a Naive POMCP. The dots denote floor cells through which participants can move. Red dots denote structural unit boundaries.

## 2 METHODS

### 2.1 STRUCTURED SPATIAL DOMAIN

We examine people's planning strategies by adapting a version of Maze Search Task (MST) previously used to study human behavior in spatial navigation domains (Kryven et al., 2024; 2021; Geva-Sagiv et al., 2025). The objective of MST is to navigate a series of partially observable, two-dimensional grid-worlds, finding exits hidden in each. Each environment has only one exit. The environment are partially observable, with the exits initially placed at a random unobserved location (black cells). Fig.2 shows a simple MST environment seen by participants during one of the practice trails. [1]. Full experiment instructions are given in Appendix I. People navigate by using their keyboard keys to move to any unoccupied grid cells adjacent to their character (a round avatar). The black hidden cells are revealed when they come into the avatar's line of sight. When revealed, the exit becomes visible as a red tile. As soon as the character moves over the exit, the trial ends. In our adaptation of MST all mazes are structured, and contained between 2 and 20 repeating structural units. The units may have occurred as reflected or rotated instances, where the structured area comprised between 80 - 100% of the environment layout.

### 2.2 COMPUTATIONAL MODELS

Decision making under partial observability can be modeled by a partially observable Markov decision process (POMDP). Equivalently, it can be viewed as a fully observable search through a space of beliefs, where each belief is a probability distribution over possible states. Solving POMDPs is notoriously hard (Madani et al., 2003), hence understanding how people approach these problems holds deep importance for cognitive science and AI.

Formally, a POMDP is a tuple $\langle \Delta(S), A, \tau, r, b_0, \gamma \rangle$, where $\Delta(S)$ is the space of probability distributions over a state space $S$, $A$ is the set of actions, $\tau$ is the belief update function, $r$ is the reward function, $b_0$ is the initial belief, and $\gamma$ is the discount factor. The belief state evolves deterministically via $\tau$, reflecting both the agent's actions and observations.

In this work, each state $s \in S$ is represented as an $N \times M$ grid whose cells are labeled {wall, empty, exit, agent}. The overall state space $S$ consists of all such grids containing exactly one agent and one exit. A belief $b \in \Delta(S)$ is thus a probability distribution over these grids, encoding the agent's uncertainty about the true state. Initially, $b_0$ assumes that the agent and the walls

---

[1]A demo is available here: `http://18.25.132.241/fragments/int_exp.php`

are known, while the exit is uniformly distributed over all valid, unseen cells. The action space $A$ contains four possible movements (up, down, left, right). Observations $o \in O$ reveal the visible subset of the grid around the agent, with each visible cell labeled {wall, empty, exit}, and any cell outside the agent's visibility range $r$ labeled as *unseen*. Observations are consistent with the grid structure of the true state $s \in S$.

The belief update function $\tau$ is given by

$$b'(s') \;\propto\; Z(o \mid s') \sum_{s \in S} T(s', a, s)\, b(s),$$

where $T(s', a, s)$ is the transition function, and $Z(o \mid s')$ is the observation likelihood. The transition function $T(s', a, s)$ specifies the probability of transitioning to state $s'$ from $s$ after executing action $a$. Here, actions that would move the agent into a wall result in the agent remaining in its current position, and transitions to an exit state terminate the process. The observation function $Z(o \mid s')$ encodes the likelihood of observing $o$ given $s'$, where observations reflect the visible subset of the grid within range $r$ of the agent's position. Visibility is blocked by walls, such that cells beyond a wall are labeled as *unseen*. Finally, the reward function $r(b, a)$ is the expected reward under the belief $b$. Since the agent can always see an exit before reaching it, $r(b, a) = 1$ if action $a$ leads the agent to a known exit and 0 otherwise.

**Expected Utility**  The optimal policy for this POMDP can be found through a belief space tree search (Kaelbling et al., 1998). The search is conducted over a tree where each node represents a belief $b \in \Delta(S)$, and edges correspond to action-observation pairs $(a, o)$. Starting from the root node $b_0$, the tree expands by simulating actions $a \in A$ and updating beliefs using the belief update function $\tau$. For each action $a$, the agent considers all possible observations $o \in O$, with the likelihood of each observation determined by the observation function $Z(o \mid s')$. At each node, the value of a belief is computed recursively using the Bellman equation:

$$V(b) = \max_{a \in A} \left[ r(b, a) + \gamma \sum_{o \in O} P(o \mid b, a) V(\tau(b, a, o)) \right], \tag{1}$$

where $P(o \mid b, a)$ is the probability of receiving observation $o$ after taking action $a$ under belief $b$. The optimal policy $\pi^*$ is derived by selecting the action at each belief node that maximizes the expected value. See (Kryven et al., 2024) for further details on this implementation, which was used as a model of human planning in MST in prior work.

Although this is the optimal strategy, human behavior has previously been shown to diverge at times from its predictions (Kryven et al., 2024), where the extent of this divergence varies between individuals in a way that can be explained by the amount of cognitive resources people allocate to planning (Kryven et al., 2021). Previous work with MST, as well as with related non-spatial planning tasks (Huys et al., 2015), has found that people's divergence from the optimal trajectories is most readily explained by a limited planning horizon ( discount factor $\gamma < 1$ in Equation 1). In the remainder of this section we describe alternative computational hypotheses for how humans could make decisions in this environment by reasoning about structural patterns.

**Generative Structure-Based Framework (GenPlan)**  Next, we describe a modeling framework that formalizes planning strategies conditioned on automatically discovered latent structure of the state-space. Our model consists of two modules: a Generative Map Module (GMM) and Structure-Based Planner (SBP). See Fig.3 for a high-level overview of this architecture. The GMM recovers a programmatic representation of the observed state-space as a composition of structural units. The SBP then uses a planner to plan a piece-wise policy once per-unit, in contrast to a global policy, saving computing costs. Importantly, this reconstructed programmatic representation is a cognitively-inspired state-space compression. While such a reconstruction may match the ground-truth planning state-space, it does not need to be exact as long as it is sufficient to serve the agent's goals (Ho et al., 2022). In theory, the cognitive principle of combining automatic structure discovery with structure-aware planners can apply to any domain, as a proof of concept here we focus on spatial tasks.

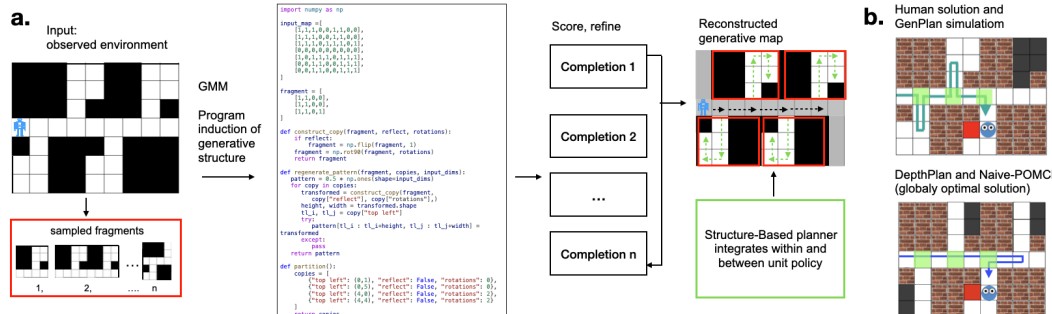

Figure 3: Model Architecture and predictions. (a) The GenPlan Framework. GMM recovers a small distribution over generative maps based on input (a partial observation of the ground truth map). Each generative map constitutes a set of candidate units coupled with a program for reconstructing the input from them. We assign a likelihood to each generative map, and pass the most likely reconstruction to SBP. SBP plans a policy once for each structural unit. (b) A human solution to the given example agrees with prediction of GenPlan (top panel). Arrows indicate the participant's path. The path predicted by alternative models DepthPlan and Naive-POMCP (bottom panel) instead optimizes the global policy. Green highlighting shows discriminating decisions.

**Generative Map Module (GMM)**   Let $I$ be the global partially observed input grid map of cells $\mathcal{S}_I$. Here, we assume that $I$ specifies all wall locations, but does not reveal the reward location, matching the information given to people by MST design. In the general case, $I$ can extend to any partial observation, as GMM will attempt to infer a structural unit from any input its given.

The GMM implements approximate inference of a posterior distribution $p(\mathcal{M}|I)$ over cognitive maps $\mathcal{M}$ that partition $I$ into structural units $\mathcal{M} = \{U_i\}_{i=1}^{m}$. We use LLM-based program synthesis with gpt-4, chosen for its strong code synthesis capabilities, to search for programs that generate $\mathcal{M}$ based on $I$ (Fig. 3). In Appendix F, we show that our approach can generalize to other LLM architectures, as long as a given LLM is able to infer at least one good-enough map in the distribution it infers. To do this, we prompt LLM to identify repeating units in the input map, and synthesize a Python program that approximately reconstructs the input from them. Prompts are given in Appendix G. The prompt includes Python code with functions describing admissible transformations, as well as a likelihood function for a given $\mathcal{M}$. In our implementation the input map $I$ is a grid-world, specified by an numerical array, where each grid cell is associated with a number (e.g. wall=1, floor=0). The reconstructions $\mathcal{M}$ do not allow overlapping units, and allow any units that repeat in $I$ at least twice.

To develop a space of possible map representations, we estimate the likelihood of each candidate $\mathcal{M}$ by a weighted combination of grid-level similarity, a function of total information in a candidate unit, and the Minimum Description Length (MDL) principle (Rissanen, 1978). MDL penalizes each unit occurrence by the bits necessary to specify the map reconstruction: their locations (effectively, the number of copies), rotations and reflections. This means that reconstructions made up of many smaller pieces are less likely than reconstructions made up of bigger ones. The function of total information in a unit ensures that the selected units are neither trivial (e.g., uniform blocks of cells made up of either walls or open space) nor noise, by defining an inverted-U relationship between likelihood and informativeness (Kidd et al., 2012). This treats large units where perception is subject to information bottleneck constraints as less likely (Cheyette & Piantadosi, 2020). Overall, this likelihood function can express a weighted preference for (1) more accurate reconstructions, (2) simpler units (to reduce planning cost).

To form the posterior $p(\mathcal{M}|I)$, we use the likelihood:

$$l(\mathcal{M}; I) \propto \frac{w_1}{d_x \cdot d_y} \sum_{x=1}^{d_x} \sum_{y=1}^{d_y} \left(I(x,y) - O(x,y)\right)^2 - w_2|\mathcal{M}| + w_3 \exp\left(-\frac{(H_{\text{total}} - \beta)^2}{2\sigma^2}\right), \quad (2)$$

where $d_x, d_y$ are input dimensions, $O$ is the output (reconstructed map), and $w_i$ are weights associated with reconstruction accuracy, map complexity, and unit complexity – free parameters of

the model. Here, weights for map complexity and unit complexity directly control planning cost, as smaller units are less costly to plan in. The expression $w_2|\mathcal{M}|$ controls the number of units (map complexity), meaning that a lower $w_2$ corresponds to a lower planning cost. The expression at $w_3$ controls total entropy in the unit (unit complexity), where more complex units incur higher planning costs. Therefore, a lower $w_3$ corresponds to a lower planning cost. See also Appendix E for the discussion of these parameters. We define the total entropy in the unit as $H_{\text{total}} = d_x d_y \cdot (-p \log_2 p - (1 - p) \log_2(1 - p))$, where $p$ is the fraction of 1's in the array, and free parameter $\beta$ reflects the processing bottleneck, in line with perceptual models explored in prior work (Cheyette & Piantadosi, 2020; Kryven et al., 2024). This definition ensures that the information term next to $w_3$ reaches a maximum of 1 $H_{\text{total}} = \beta$, but decays to 0 on both sides of this maximum. In the general case, input $I$ and output $O$ are real-valued 2D image arrays. In our current implementation input $I$ takes values 0 and 1, and the output $0 \leq O(x, y) \leq 1$. Instead of using the raw Python program for $\lambda$ to measure its complexity, we use a compressed encoding of the units and their transformations used to reconstruct the map. Here, compressing LLM-generated output and transformations is analogous to refactoring the synthesized programs. As the length of LLM-synthesized code may be noisy, due to injected comments and code redundancies, refactoring the output obtains a denoised metric of complexity.

The posterior $p(\mathcal{M}|I)$ defines how the environment structure is encoded into memory. The free parameters $w_i, \beta$ balance reconstruction accuracy against the complexity of the generative structure and planning costs. Fig.3 shows a simple example with structural units highlighted in red. Bigger examples designed in our simulation experiment are shown in Fig.2. Our proof-of-concept implementation uses the most likely generative map in the SBP module for generating a policy. This approach makes a simplifying assumption relative to prior work on human cognitive maps (Sharma et al., 2022), which found that people anticipate unseen map structure by maintaining a distribution over possible maps. However, as our experiment design does not distinguish between planning over a distribution or the most likely map, both approaches would make identical predictions.

**Structure-Based Planner (SBP)**  Implementing SBP integrates planning within and between structural units. Since finding an exact solution is intractable due to the size of the problem (Kaelbling et al., 1998), we solve planning *within a unit* by searching through the belief space using an approximate online Partially Observable Monte Carlo Planner (POMCP) (Silver & Veness, 2010). A plan *in-between units* consists of leaving the current unit and transitioning to the next one. We solve the former by adapting Monte Carlo Tree Search (MCTS), and introducing an optimistic heuristic valuation for open cells around the boundary of a unit (i.e. cells through which we can exit the unit). Upon completing a plan for the current unit, this heuristic should encourage us to leave the unit in the direction that minimizes the expected global cost of reaching the exit. In the general case, this can be any heuristic that does not overestimate the true cost. Here, we compute the values of boundary cells as inversely proportional to the average of manhattan distances to the remaining external unobserved cells, hence assigning a higher value to cells that are on average closer to the remaining unseen parts of the map. We solve the transition to the next unit using a POMCP on the global map, but implement the option to switch to within-unit planning upon reaching the new unit. Here, we evaluate the option by estimating the average per-step cost to plan within the unit. The pseudo-code for GMM and SBP the algorithms is given in Appendix H. In Appendix B we analyze SBP by deriving worst-case bounds on step-cost differences between SBP and the optimal policy. We show that SBP yields only a constant-factor step penalty that does not affect asymptotic exploration, and in non-structured settings the worst-case costs are identical.

## 2.3 COMPUTATIONAL HYPOTHESES

We compare human performance to the hypotheses (planing algorithms) to evaluate whether and how human planning implements the two computational steps outlined by GenPlan.

1. **Structure-Naive Planner (Naive-POMCP)**: The model doesn't use generative maps, and plans by optimizing a global policy for the environment.

2. **Structure-Naive Planner With Cognitive Constraints (DepthPlan)** The model doesn't use generative maps, and plans by optimizing a global policy that discounts future states to model limited planning depth. This model was previously used to describe how people plan in MST (Kryven et al., 2024).

3. **A Generative Planner (Gen-POMCP)**: The model uses generative maps, and plans a reusable policy based on the most likely map from the distribution of induced maps $p(\mathcal{M}|I)$ (see Fig. 3a and b).

Naive-POMCP uses an approximate POMCP planner designed for large partially-observable state-spaces. It was previously used to model human planning in large domains, although this work did not examine the effect of environment structure on human plans (Sharma et al., 2022). DepthPlan is consistent with prior work that models human deviations from optimal cost minimization by limited planning horizon, without modeling environment structure (Kryven et al., 2024; Huys et al., 2015). Since DepthPlan internally computes an optimal policy, it can only be applied to smaller environments (generally 6 or fewer structural units, see Experiment 1). Only the third hypothesis is consistent with the two computational steps proposed by the GenPlan framework: it represents maps as generative programs, and uses this structure to plan a reusable policy, reducing planning costs.

## 3 EXPERIMENTS

We first test whether people use structure to reduce computational costs of planning, as implemented by Gen-POMCP, in contrast to DepthPlan – the state-of-the-art model of planning in MST (Kryven et al., 2024) (Experiment 1). For this experiment, we use a set of 20 structured environments at a scale that can be solved by DepthPlan, in order to compare DepthPlan and Gen-POMCP. Based on the sample sizes used in a previous study of human planning in structured spaces Sharma et al. (2022), and a preliminary pilot showing a strong effect of structure on planning, we recruited a N=30 participants as sufficient for confirm this effect. We then run a simulation experiment with 10 large environments containing 20-25 units (Experiment 2) to compare the computational costs of Gen-POMCP and Naive-POMCP, demonstrating that Gen-POMCP requires significantly fewer computational resources.

### 3.1 EXPERIMENT 1: BEHAVIORAL VALIDATION

**Procedure**  The experiment was conducted in a web browser, using the web-interface of MST (Kryven et al., 2024). Before beginning the experiment participant gave informed consent and completed a series of practice trials, followed by an instruction quiz. Following this, they completed a variant of MST with structured mazes, with exit locations randomly chosen at the time of design. After completing the experiment, we administered a post-experiment questionnaire collecting demographic information. As our goal was to observe ecologically-valid planning, we did not offer performance-based incentives, and simply informed participants that the exit could be in any of the hidden tiles, and instructed them to find it in each environment.

We recruited 30 (13 female, 17 male, $M(age) = 36.7$, $SD(age) = 13.5$) english-speaking participants on Prolific, who were paid 9£ per hour. None were excluded. On average the experiment took 10 minutes to complete. The experiment was approved by our institution's IRB.

**Behavioral metrics**  We introduce the following *behavioral definitions* to quantify people's alignment with the GenPlan framework.

- A set of *discriminating decisions* $\mathcal{D}(I)$ in a given environment $I$ is the subset of all states in $I$ where Gen-POMCP and Depth Plan predict a different most likely action. That is, $\mathcal{D}(I)$ includes only actions diagnostic of structure-based planning. Fig. 3b illustrates discriminating decisions in a simple example. Unlike the global solution (bottom panel), Gen-POMCP and most humans search by entering inside the structural units. The cells highlighted in green are discriminating decisions.

- *Modular fraction* $\sigma(\mathcal{D})$ defines the fraction of decisions in a given set of discriminating decisions $\mathcal{D}$ that are more likely under Gen-POMCP, compared to DepthPlan. Notably, as weights $w_i$ in equation 2 can tradeoff accuracy against representation and planning costs, Gen-POMCP can capture flexible strategies that integrate global and local search. For simplicity, here we assume stable population-level weights that strongly favor structure over accuracy, meaning that $\sigma(\mathcal{D})$ is a conservative estimate of how well Gen-POMCP explain human behavior.

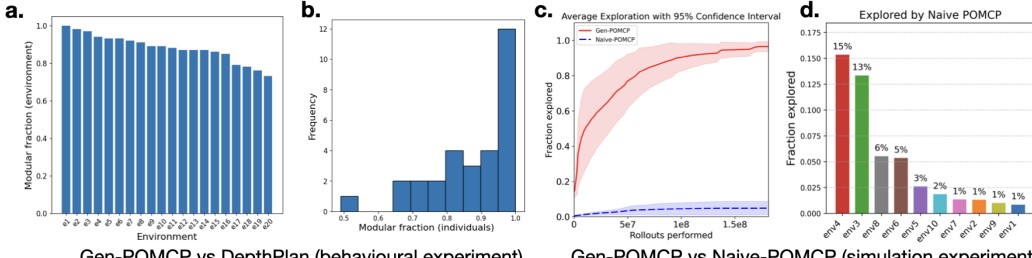

Figure 4: Experiment result shows that Gen-POMCP explains people better than DepthPlan. (a.) Modular fraction for each environment (b.) The histogram of individual modular fractions per participant, over all discriminating decisions. (c) The fraction of each environment explored by Gen-POMCP across environments (solid line) and Naive-POMCP (dashed lines) as a function of compute budget (number of MCTS rollouts). (d) The fraction of each environment explored by Naive-POMCP, if given the amount of MCTS rollouts at which Gen-POMCP fully explores the given environment. Each bar shows a different environment.

**Results** Our results reveal that Gen-POMCP predicts human behavior significantly better than DepthPlan (Fig. 4). Examination of modular fractions for individuals and environments shows that people are highly consistent with our model, demonstrating structure-based planning across all environments and individuals. Across all environments, human behavior is better explained by Gen-POMCP, suggesting that people approximate planning by piecewise policies conditioned on structure, rather than by planning a global policy with a limited depth. Fig. 3b illustrates the difference between the models in a simple example. Like Gen-POMCP, the majority of people search the environment by a trajectory shown in the top panel. However, both the optimal policy predicted by the Naive-POMCP, and the depth-limited planning as implemented by DepthPlan, predict the trajectory shown at the bottom – because it allows to quickly reveal a large portion of the global map.

### 3.2 EXPERIMENT 2: SIMULATION

Next, we compare the computing resources needed by Gen-POMCP and Naive-POMCP to explore 10 large structured environments (e.g., see Fig. 2c). As the objective of MST is to find a hidden exit, each simulation is set up to run until the environment is fully explored (i.e., the exit is not revealed until the end of the simulation). We compare two quantitative comparisons. For each environment we compute the fraction of the environment that each model is able to explore given a certain compute budget (Fig. 4d), finding that the Gen-POMCP required a much smaller budget to search the entire environment. Fig.4c separately shows the fraction of each environment that Naive-POMCP is able to explore, if given the rollout budget at which Gen-POMCP has searched the entire environment. In AppendixA we give a proof that the length of the search trajectories produced by Gen-POMCP exceeds trajectories produced by Naive-POMCP by a bounded amount, demonstrating that Gen-POMCP also produces good enough plans. All environments used in Experiments 1 and 2 are shown in AppendixB.

## 4 RELATED WORK

**Models of Human Planning** People make near-optimal plans in natural domains, such as city navigation (Bongiorno et al., 2021), yet often perform sub-optimally in laboratory-based behavioral paradigms such as multi-arm bandits (Keramati et al., 2016; Huys et al., 2015), strategic games (Ferreira, 2013), and sequential decision-making (Unterrainer et al., 2004; Kryven et al., 2024; Callaway et al., 2022). Such deviations from optimality are often explained by approximate planning with a limited planning horizon (Ferreira, 2013; Kryven et al., 2024; van Opheusden et al., 2023). Recent work (Correa et al., 2025) examines how people represent policies in programmatic forms that minimize description lengths, finding sensitivity to both effort minimization (similar to seeking shorter search paths) and MDL (shorter programs). Similar to our work, their study found that human plans

heavily favor reuse. Unlike our work, experimental paradigms used in these studies lack the regular problem-domain structure ubiquitous in the real-world.

**Cognitive Maps**   A recent study found that people form cognitive maps that facilitate planning (Ho et al., 2022), by selectively representing only the goal-relevant parts of the map. Like our work, this study assumes that human cognitive maps are learned by compressing observations. Unlike our work, this study uses unstructured maps. A recent study of exploration in structured environments found that people anticipate environment structure, even when not informed about them in advance (Sharma et al., 2022), and can predict unseen parts of the map. Unlike our work, this study focuses solely on map prediction, and does not examine the role of cognitive maps in planning. It also relies on exhaustive enumerative search to discover the underlying map representations, unlike our work that implements tractable inference using LLM-based program synthesis.

**Hierarchical Reinforcement Learning (HRL)**   Reinforcement learning methods reduce planning complexity through hierarchical abstractions such as options (Sutton et al., 1999), shared structure across related MDPs (Wilson et al., 2012), and predictive state merging (Singh et al., 2012). Classical approaches include abstraction spaces (Sacerdoti, 1974), HTNs (Erol, 1995; Nau et al., 1999), and probabilistic propositional planning (Littman, 1997), while recent work analyzes state abstraction in tree search (Anand et al., 2016; Hostetler et al., 2014; Hutter, 2016) and structural conditions for efficient planning (Wen et al., 2020). Related lines study option discovery (Jinnai et al., 2019; Ivanov et al., 2025), generalized planning across task families (Curtis et al., 2022), and symmetry-based representations (Silver et al., 2017). These approaches engineer abstractions to improve worst-case or average efficiency. In contrast, GenPlan conditions planning on inferred structure: the GMM recovers repeated fragments from partial observations, and SBP reuses fragment-level policies with a focus is on explaining human planning behavior. While comprehensive theory of asymptotic guarantees for of GenPlan is outside the scope of our work[2], in Appendix B we show that the worst-case performance of GenPan differs from optimal POMCP only by a constant factor.

**Using LLM to plan**   Several related works have used LLM for offline planing. Similar to our work, Parsel (Zelikman et al., 2023) leverages LLMs to decompose complex tasks into modular components that can be composed to solve a larger problem. Unlike our work, Parsel solves problems specified in natural language, rather than using LLMs to infer latent environmental structure for cognitive maps. A study of Kim et al. (2024) proposes a framework that fine-tunes LLM-based agents to plan in grid environments by constructing an internal representation. Unlike GbenPLan, which represents cognitive maps in code, their model works with text-based grid-world descriptions and uses unstructured maps. Chain-of-Thought Procedure Cloning enables agents to generalize to new fully observable environment configurations by imitating the intermediate reasoning steps of expert procedures (Yang et al., 2022). The ReAct framework (Yao et al., 2022) guides anLLM through iterative thought-action steps, that can apply to spatial navigation domains. However, ReAct does not provide inference over enironmental structure, or mechanisms for policy reuse (Gu et al., 2024). Unlike GenPlan, these models do not target planning in structured spatial environments, and are not evaluated in human experiments.

**Natural priors.**   Adaptive real-world planning draws on complex prior knowledge of the world (Acquaviva et al., 2022; Spelke & Kinzler, 2007; Dehaene et al., 2006). Learning natural priors that make people so efficient in real-world remains an important problem in cognitive AI (Kumar et al., 2022; Li et al., 2024; Binz et al., 2024). (Feldman, 2013). Similar to our work, an emerging line of research leverages LLMs as a back-end to planning frameworks as a way of informing planning by the implicit natural priors embedded in LLM though training on vast amounts of human data (Tang et al., 2024; Correa et al., 2025; Towers et al., 2024; Xie et al., 2023; Piriyakulkij et al., 2025; Curtis et al., 2025). Our computational framework builds on this approach, focusing on modeling how cognitive maps and planning policies may be learned together.

---

[2]For example, it may be possible to adapt computational efficiency guarantees from Wen et al. (2020) to our setting.

## 5 DISCUSSION

We give a computational account of how people plan in structured environments by integrating (i) generative reconstruction of compressed cognitive maps from observations, coupled with (ii) structure-conditioned policy reuse. GenPlan operationalizes these two principles in a computational framework that (1) represents maps by approximate generative programs in a Turing-complete language, and (2) plans in these representations using POMCP with policy reuse. We adapt an experimental Maze Search task previously used to study human planning, to show that in structured environments human planning is consistent with these two computational principles, in contrast to the state-of-the art model of depth-limited planning proposed in previous work. This result makes a *scientific contribution* by showing that environmental structure influences the selection of planning strategies: human deviations from optimal policies are, at least in part, due to approximating planning by piecewise policies conditioned on structure for policy reuse. Our GenPlan framework makes an *engineering contribution* by showing how to actually build these principles into a working system. GenPlan is compute-efficient, because it achieves tractable inference over the underlying map structure by leveraging LLM-driven program generation (in contrast to enumerative search e.g. (Veness et al., 2011) ) and because it reduces the amount of planning compute through policy reuse.

We note that in our experiments planning varies between individuals, in line with variability observed in previous work Callaway et al. (2022). Our model proposes a computational account explaining this variability as arising from how individuals may represent the same map in different ways. For example, depending on available cognitive resources, someone may form representations made up of coarser or finer patterns. In the limit, GenPlan can prioritize reconstruction accuracy and treat the entire environment as a single structural unit, solved by a globally optimal policy. Our implementation makes a simplifying assumption that that reconstruction and description weights are fixed and stable at population level, which works well to explain behavior in our experiment. Future work can further consider the stability and generalization of these parameters within and between individuals. General case solutions can be built to account for flexible cognitive resources, allowing the model to switch between map representations in response to changing cognitive demands. Future work can also examine how the weights should be chosen to optimally balance the computational costs of planning and memory, against utility.

**Limitations and future work.** Our choice of MST environment is motivated by prior state-of-the-art model on human spatial planning (Kryven et al., 2024). In alignment with prior work on human spatial planning (Kryven et al., 2024; Ho et al., 2022; Sharma et al., 2022), GenPlan relies on deterministic units that are stable over time, which allows comparison of our results with prior work. As natural environments are often probabilistic and evolving, future work should examine building blocks that differ superficially (e.g. square or rectangular city blocks), while preserving probabilistic generative constraints. To work with probabilistic units, GenPLan would rely on the same two congitive principles of structure-based map compression and policy reuse, while using a different implementation of GMM and SBP where the unit maps themselves are specified by generative programs in a graph-grammar or a probabilistic CFG. We hope that our results will inspire future work on human planning in structured environments beyond grid-world domains (e.g. music performance, reasoning in cognitive graph domains).

Another promising direction of future work consists of modeling how different goals shape cognitive maps. For instance, people tend to perceive San Francisco as having a grid layout, even though its map reveals a more complex structure. Such grid-like intuitions could arise from goal-dependent cognitive maps (Ho et al., 2022) – where a local grid model may actually be good enough to plan a pedestrian shortcut across a neighborhood. Because our aim is to explain behavior, we do not provide full theoretical evaluation in terms of search efficiency, completeness, and scalability to arbitrary task sets. We hope the principles we proposed will motivate further theoretical work on when human-like priors are advantageous or suboptimal.

**Implications.** While planning cognition has been studied extensively, human planning real-world planning domains remains underexplored. A computational-level understanding of how human planning adapts to real-world environments, given their distinctive properties such as structural and inductive biases, can inform models that not only empower AI to better understand and assist humans, but also decrease environmental impact of AI algorithms, by enabling them to achieve effective policies with less compute. In contributing a proof of concept implementation, GenPlan brings new insights into human spatial planning, and takes a step toward building cost-efficient planning in AI.

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

## A CODE AVAILABILITY

The Gen-POMCP implementation is available here: `https://anonymous.4open.science/r/GenPlan-FBCD/README.md`

## B PERFORMANCE BOUNDS FOR STRUCTURE-BASED PLANNING

In structured environments Gen-POMCP can explore the environment faster than Naive-POMCP (using fewer rollouts and in less time) by taking advantage of limited resources. However, it simplifies the planning problem by entirely exploring each fragment it enters before moving to the next. This heuristic can result in longer overall paths taken to search the environments. It is reasonable to ask by how much the global Naive-POMCP can actually improve on the path length taken by Gen-POMCP (and specifically the Structure-Based Planner).

Below we sketch a proof that considers the limit in which each planner fully optimizes its respective objective: Naive-POMCP follows the Bayes-optimal plan in each fragment and Gen-POMCP follows the Bayes-optimal global policy for the maze. We bound the cost difference according to the worst-case cost in steps.

**Expected and worst-case** The expected number of steps it takes for a policy to explore a maze is the average over the length of path this policy takes to reach uniformly sampled exit locations. The worst-case number of steps is the largest number of steps that the policy could take for some exit position. This is bounded below by the number of steps required to fully explore the maze.

**Lemma 1.** *There exists a fragment of size $n \times n$ which takes $O(n^2)$ steps to search in expectation, and to explore fully.*

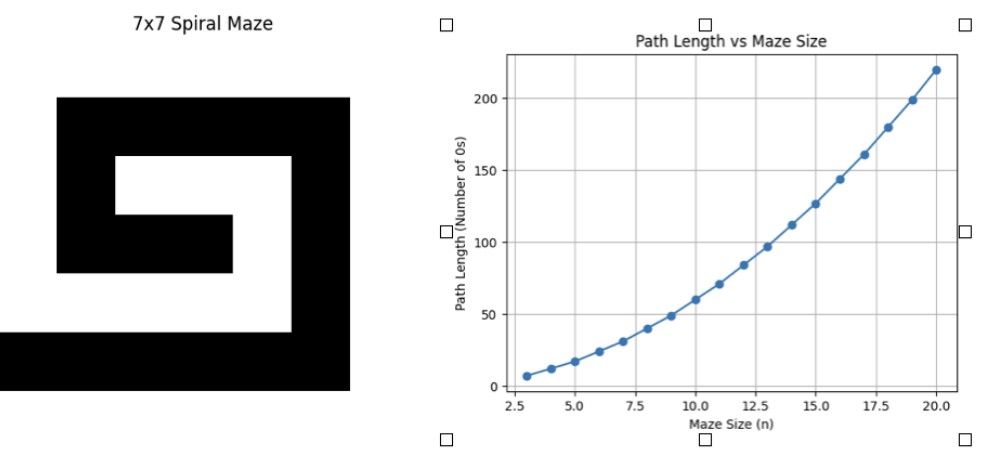

Figure 5: Consider a maze with a spiral wall - the white cells indicate traversable floor, and black indicate intraversable wall. Simulating these environments shows that the maximal length of path (white cells) in the environment grows as $\approx \frac{1}{2}n^2$
.

*Proof.* Consider a fragment with the maximum spiral path (e.g. Figure 5). The length of this path scales quadratically with $n$. In particular, following a spiral path takes a series of four legs at each depth, and the length of every other leg reduces by two (one for the wall and one for the path itself).

This yields

$$n + \sum_{i=0}^{\lfloor n/2 \rfloor - 1} 2(n - 2i - 1) = \frac{n}{2}2n - 4\frac{(n/2)(n/2+1)}{2} + O(n) \qquad (3)$$
$$= \frac{1}{2}n^2 + O(n)$$

$\square$

**Theorem 2.** *In the an $n \times n$ maze, the expected number of steps taken by SBP may exceed the expected number of steps of an optimal policy by $\Omega(n^2)$.*

*Proof.* Build a fragment by adjoining an empty room and a spiral by a single door at a corner. Now connect the two fragments by adding a door between the empty rooms in the opposite corner. Assume the size of the empty rooms is such that the optimal algorithm can find the exit with probability $1/2$ by checking each empty room, but the SBP algorithm must explore entirely the first fragment that it enters. With probability $3/4$, the exit is not in the first empty room, so it must explore the spiral, which takes time $\Omega(n^2)$ to fully explore by Lemma 1. The spiral also must be exited, so around $n^2$ steps are spent when the exit is in the other empty room (in this case the optimal planner finds immediately by checking each room). Since the optimal planner takes only a constant number of steps to check each empty room, and then behaves identically to the SBP, the expected cost when the exit is in any other location is asymptotically the same, so the expected cost difference is roughly $\frac{1}{4}n^2 = \Omega(n^2)$. $\square$

**Theorem 3.** *The number of steps to fully explore a maze is $O(n^2)$.*

*Proof.* Consider a $v$-vertex connected graph. The maximum width (roughly achievable by the spiral) is $v$, leading to a naive bound of $O(v^2) = O(n)$. This can be improved to $O(v)$ by running a depth-first search. Since there are 4 movement directions the degree of this graph is 4 meaning the maximum number of backtracks *to* a vertex is 3, which immediately gives $4v$. However, in a depth first search there is only one backtrack *from* each vertex is 1, which leads to an easy inductive proof that the bound is $O(2v - 1)$ regardless of degree, yielding $2n^2 - 1 = O(n^2)$. Note that further improvements should be possible by considering the number of walls required to induce the worst-case topology. $\square$

This implies that the Bayes-optimal policy has $O(n^2)$ expected cost (since its *expected* cost must be at least as good as the *expected* cost of exhaustive search), regardless of the maze. Together, Lemma 1 and Theorem 3 demonstrate that the SBP heuristic does not damage the (asymptotic) expected cost in the worst maze.

**Theorem 4.** *Assume that an $n \times n$ maze is fragmented in such a way that any time a fragment is entered, it can be fully explored before exiting, into $c^2$ square $(n/c) \times (n/2)$ fragments. The asymptotic expected cost is $\Theta(n^2)$ in the worst such maze for the modular optimal and globally optimal policies.*

*Proof.* First, consider the global optimal policy. The additional requirements placed on the maze cannot make the $O(n^2)$ bound in Theorem 3 worse, and we can get a matching lower bound by simply adjoining multiple spiral examples as in Lemma 1 and adding doors between them.

Now consider the modular optimal policy. It is clear that the globally optimal policy has an expected cost as least as low as the modular optimal policy (even in their respective worst mazes), by definition, so the $\Omega(n^2)$ lower bound automatically carries over to the modular optimal policy. We assumed that the modular optimal policy takes the Bayes-optimal paths between fragments. This must be at least as good as the following strategy: mimic the global optimal policy, but any time a new fragment is entered, first explore it completely and return to the entrance. By Theorem 3, each such "extra" exploration detour takes at most $2(\frac{n}{c})^2 - 1$ steps, and the return takes at most $(\frac{n}{c})^2$ steps. The total is $3(\frac{n}{c})^2 - 1$. There are exactly $c^2$ such detours, for $4n^2 - c^2 = \Theta(n^2)$ extra steps. The global optimal policy also takes $\Theta(n^2)$ steps.

$\square$

Therefore, in the worst case the modular algorithm is inferior by at least a constant factor of the total search time in expectation. Examining the proof of Theorem 4 yields a factor of 2.5 over our upper bound in Theorem 3, but presumably this can be improved substantially since a lot of exploration is being redone after the detours.

**Improving expected cost upper bounds**   Substantial improvements to the worst-case cost bound in Theorem 3 are easy to obtain when the proof is applied to expected cost by e.g. noting that the depth-first search visits at least one new cell every two steps, meaning that there is clearly at least a $1/4$ chance of finding the exit after $n^2$ steps, or by noting that the true number of "vertices" is reduced by walls. These improvements seem to apply equally to the modular and global optimal policies, and probably do not affect our constants much.

For worst-case cost, the situation is similar. However, the worst-case cost analysis simplifies significantly with the additional assumption that transitions between fragments are negligible (say, if they all branch off from a central room). This observation is trivial but worth stating explicitly:

**Theorem 5.** *When the cost to transition between fragments is negligible, each has one entrance, and there is no line-of-sight across fragments, the modular algorithm has the same worst-case step count as the optimal algorithm.*

*Proof.* In the worst case, the optimal algorithm must explore each fragment, and since there is only one entrance to each fragment it is not possible to gain any advantage by exiting a fragment before it has been fully explored. □

## C  EXPERIMENTAL ENVIRONMENTS

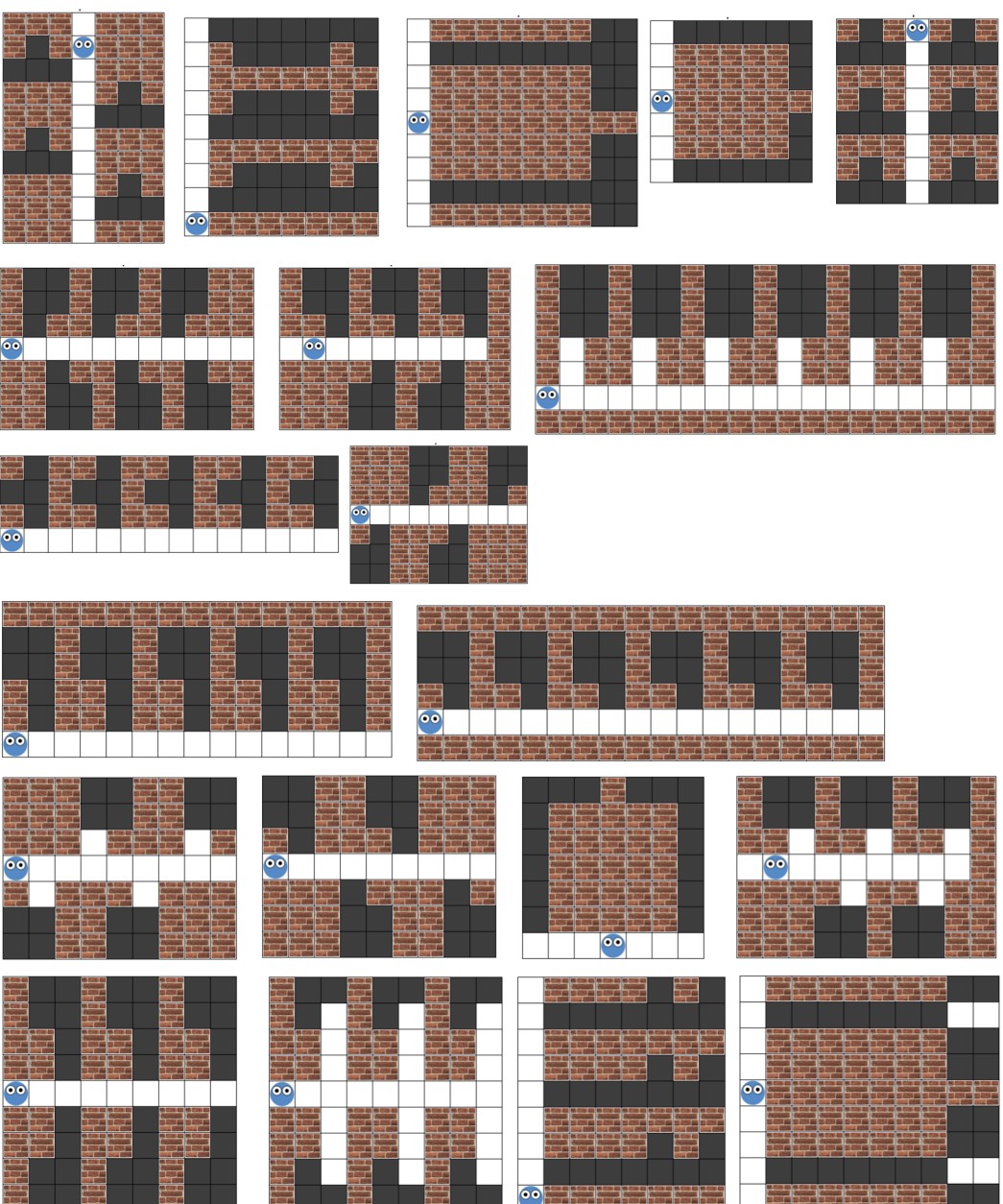

Figure 6: Environments used in Behavioral Experiment 1.

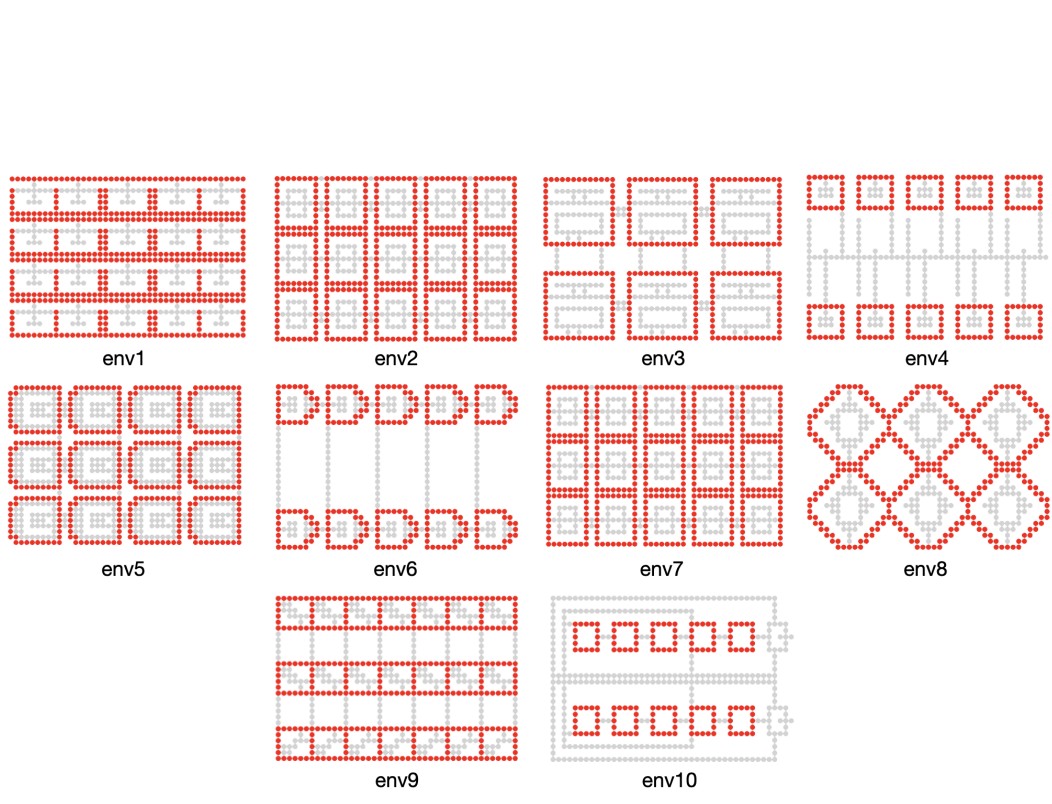

Figure 7: Environments used in Simulation Experiment 2.

# D ADDITIONAL RESULTS - SIMULATION EXPERIMENT

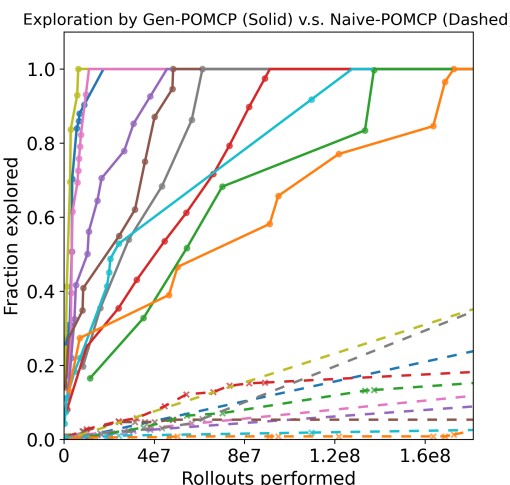

Figure 8: The fractions of each environment searched by Gen-POMCP and Naive-POMCP given identical computational budget. Gen-POMCP requires fewer rollouts and saves computing costs. Each environment is shown in a different color (see also Figure 4.)

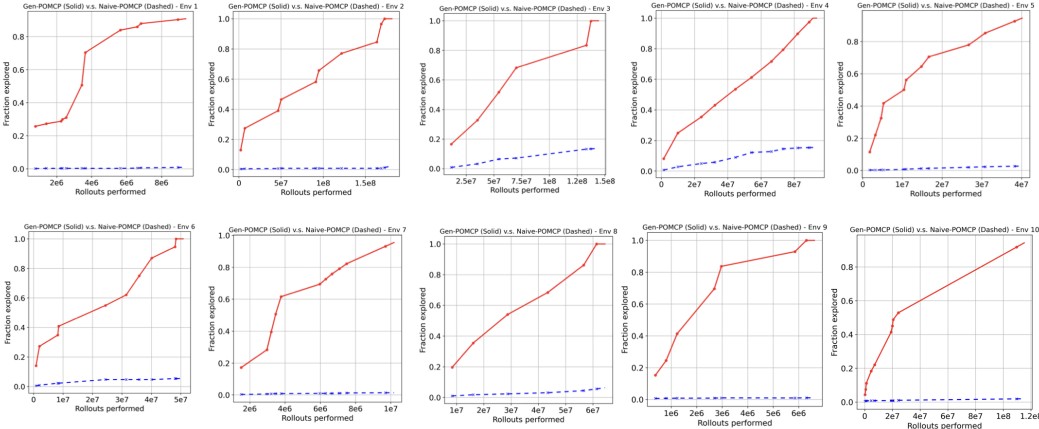

Figure 9: The fractions of each environment searched by Gen-POMCP and Naive-POMCP given identical computational budget. In each individual environment Gen-POMCP requires fewer rollouts and saves computing costs.)

# E THE EFFECT OF FREE PARAMETERS ON RESULTS

*Reconstruction accuracy.* As map accuracy decreases, the amount of online heuristic planning increases, and the amount of structure-based planning decreases. We implement this heuristic based prior work with Maze Search (Kryven et al 2024). A zero reconstruction accuracy entails a fully heuristic planning, regardless of planning cost.

*Planning cost.* As planning cost increases the units become smaller, leading to more localized search. A negligible planning cost paired with a high reconstruction accuracy reduces the model to a global planner. A high accuracy and high planning cost leads to a fully structure-based planning (the population level model used in the paper)

Dissociating between these parameters in a human experiment requires a complex targeted design, beyond the scope of the current work. As our goal is to test whether people use structure-based

planning, as opposed to global search considered in previous work, we use a population-level model with high reconstruction accuracy and low planning costs. This leads the model to plan within single units intended by design (rather than grouping them) and maximizes the amount of discriminating decisions between structure-based and global planning.

## F  GENERALIZING TO ACROSS LLM ARCHITECTURES

In the paper, we used GPT-4 to build a proof-of-concept implementation for the GMM, originally chosen due to its strong code-generation abilities. However, we clarify that the choice of LLM model and prompting strategy are not critical to our framework's results. The primary contribution of our work is showing that human planning in structured environments relies on integrating two cognitive principles – (1) compressed cognitive maps that leverage redundant structure (implemented in GMM) and (2) policy reuse (implemented in SBP).

Below we show that GMM can be implemented with different LLM architectures. To do this, we show experimental results producing similar reconstructions by using different LLMs as a backend: GPT-4, Gemini-2.5-flash, Llama-3.3-70B, and Kimi-K2-Instruct-0905. Furthermore, we present results from two different prompting strategies (one-step prompt and multi-step prompts), showing that the exact prompt wording is not critical to producing the given results.

### F.1  SINGLE PROMPT

**Input map**

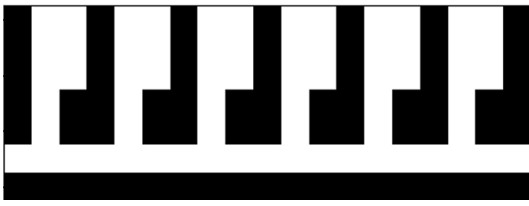

**Top scoring unit candidate**

GPT-4, Gemini-2.5-flash, Llama-3.3-70B          Kimi-K2-Instruct-0905

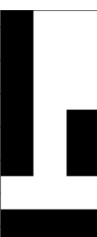          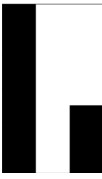

**Reconstructed map**

GPT-4, Gemini-2.5-flash, Llama-3.3-70B          Kimi-K2-Instruct-0905

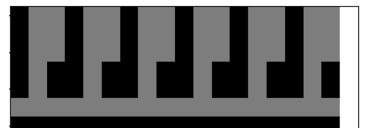          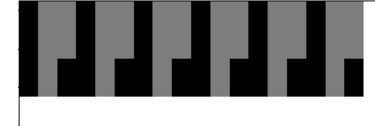

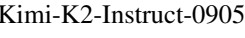

### F.2 MULTI PROMPT

**Input map**

Complete input map                                    Partial input map

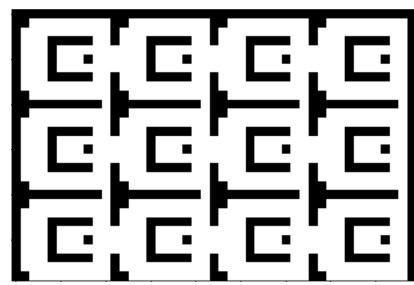                                  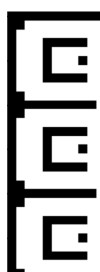

**Top scoring unit candidate**

GPT-4, Gemini-2.5-flash, Llama-3.3-70B               Kimi-K2-Instruct-0905

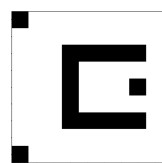                                  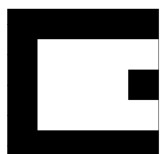

**Reconstructed map**

GPT-4, Gemini-2.5-flash, Llama-3.3-70B               Kimi-K2-Instruct-0905

                                  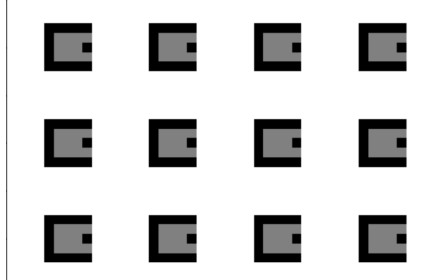

## G PROMPTS

### G.1 ONE-STEP PROMPT FOR GMM

The Single Prompt GMM identifies the unit along with the reconstruction program using one prompt to the LLM. The prompt describes the task as a two-step procedure: first identify the repeating unit, then complete and return a runnable Python program that contains both the unit as a 2D array and the reconstruction function.

System prompt:

```
You are a designer's assistant, skilled in noticing patterns,
combining fragmets into a patterns, and extrapolating them.  You
are skilled in identifying the underlying structure of a pattern
and generating new fragments that fit the pattern.  You are also
skilled at writing Python code.
```

User prompt:

```
There are two steps to this task.  In Step 1, you will be given
an input (a map) and asked to identify its constituent units.
The input is a matrix, elements of which can take values 1 and
0.  Your task is to identify a repeating unit in this input.

To be considered a repeating unit, the unit does not have to tile
the space exactly, but it must appear at least twice.  The unit
instances may be flipped horizontally or vertically, translated
horizontally or vertically, and rotated by multiples of 90 degrees
(i.e.  0, 90, 180, 270).

IMPORTANT:
1.  Instances of the unit must NOT overlap in the original input.
2.  The height and width of the unit need not be equal

Example 1.
Given input:  {example input 1}
The repeating unit is:  {example unit 1}

Example 2.
Given input:  {example input 2}
The repeating unit is:  {example unit 2}

Example 3.
Given input:  {example input 3}
The repeating unit is:  {example unit 3}

In Step 2, you will write a function that attempts to identify all
occurrences of the unit in the input.  Return a list containing
the indexical locations of the top left corner for each copy,
along with whether to reflect the copy horizontally and the
number of 90 degree counter-clockwise rotations (these operations
together generate the dihedral group D4).

For instance, in the examples above, possible solutions include

Example 1.
Solution 1:  {example 1 program 1}
Solution 2.  {example 1 program 2}

Example 2.
Solution 1:  {example 2 program 1}
Solution 2:  {example 2 program 2}
```

```
Given your unit and partition, the user will attempt to
reconstruct the input using the following function:

def construct_copy(unit, reflect, rotations):
    if reflect:
      unit = np.flip(unit, 1)
   unit = np.rot90(unit, rotations)
   return unit

def regenerate_pattern(unit, copies, input_dims):
   pattern = -1 * np.ones(shape=input_dims)
   for copy in copies:
      transformed = construct_copy(
         unit,
         copy["reflect"],
         copy["rotations"],
      )
      height, width = transformed.shape
      tl_i, tl_j = copy["top left"]
      try:
         pattern[tl_i:tl_i+height, tl_j:tl_j+width] = transformed
      except:
         pass
   return pattern

We can test the success of this regeneration with

input_map = np.array(input_map)
output = regenerate_pattern(
   unit,
   partition(),
   input_map.shape,
)

Now is your turn.  Propose a unit that can be used to reconstruct
the given input.  Respond by completing the following Python code:

input_map = {input map}
# make sure to define all arrays as numpy arrays
import numpy as np
input_map = np.array(input_map)

unit = [ ...  ]
unit = np.array(unit)

def partition():
   copies = []
   # Place your code here.  Let's think step by step
   return copies

Please include only code in you response, no text.'''
```

## G.2 MULTI-PROMPT GMM

To improve GMM scalability on large maps, we introduce a two-step approach. The two steps are
implemented in two separate prompts, which adapt the strategy described in the previous section.
The first prompt provides a part of the map and asks to identify a repeating unit. The second prompt

asks the LLM to infer a reconstruction program for the complete map given the previously identified unit.

System prompt:

```
You are a designer's assistant, skilled in noticing patterns,
combining fragments into a patterns, and extrapolating them.  You
are skilled in identifying the underlying structure of a pattern
and generating new fragments that fit the pattern.  You are also
skilled at writing Python code.
```

Unit Identification Prompt:

```
You will be given an input (a map) and asked to identify its
constituent units.  The input is a matrix, elements of which can
take values 1 and 0.  Your task is to identify a repeating unit in
this input, and ONLY output the unit as a 2D python array.  DO NOT
include anything else in the completion.

To be considered a repeating unit, the unit does not have to tile
the space exactly, but it must appear at least twice.  The unit
instances may be flipped horizontally or vertically, translated
horizontally or vertically, and rotated by multiples of 90 degrees
(i.e.  0, 90, 180, 270).

IMPORTANT:
1.  Instances of the unit must NOT overlap in the original input.
2.  The height and width of the unit need not be equal

Example 1.
Given input:  {example input 1}
The repeating unit is:  {example unit 1}

Example 2.
Given input:  {example input 2}
The repeating unit is:  {example unit 2}

Example 3.
Given input:  {example input 3}
The repeating unit is:  {example unit 3}

The input you are working with is the following map:  {input map}
```

Reconstruction Program Prompt:

```
You will write a function that attempts to identify all
non-overlapping occurrences of the unit in the input.  Return a
list containing the indexical locations of the top left corner
for each copy, along with whether to reflect the copy horizontally
and the number of 90 degree counter-clockwise rotations (these
operations together generate the dihedral group D4).

Example 1.
Given input:  {example input 1}
The unit is:  {example unit 1}
A solution is:  {example 1 program 1}
An alternative, more structured solution is:  {example 1 program
2}

Given input:  {example input 2}
The unit is:  {example unit 2}
A solution is:  {example 2 program 1}
An alternative, more structured solution is:  {example 2 program
2}

Given your unit and partition, the user will attempt to
reconstruct the input using the following function:

def construct_copy(unit, reflect, rotations):
    if reflect:
      unit = np.flip(unit, 1)
    unit = np.rot90(unit, rotations)                               )
    return unit

def regenerate_pattern(unit, copies, input_dims):
    pattern = -1 * np.ones(shape=input_dims)
    for copy in copies:
        transformed = construct_copy(
            unit,
            copy["reflect"],
            copy["rotations"],
        )
        height, width = transformed.shape
        tl_i, tl_j = copy["top left"]
        try:
            pattern[tl_i:tl_i+height, tl_j:tl_j+width] = transformed
        except:
            pass
    return pattern

We can test the success of this regeneration with

input_map = np.array(input_map)
output = regenerate_pattern(
    unit,
    partition(),
    input_map.shape,
)
```

```
Now is your turn.  Respond by completing the following Python
code,
1.  include everything that is between START OF CODE and END OF
CODE
2.  include the entire input map provided, do not use ...  to omit
3.  ONLY fill partition(), do not use variables/functions that are
not defined
4.  In the returned copies, follow the exact key names in the
examples:  'top left', 'reflect', 'rotations'.
4.  DO NOT include anything else in the completion

# START OF CODE, make sure to define all arrays as numpy arrays
import numpy as np

input_map = {input map}
unit = {input unit}
input_map = np.array(input_map)
unit = np.array(unit)

def partition():
    copies = []
    # Place your code here.  Let's think step by step
    return copies
result = partition()

# END OF CODE
```

### G.3 IN-PROMPT EXAMPLES

**Example 1**

Input map                                        Unit

```
[                                                [
  [1, 0, 1, 0],                                    [1, 0],
  [0, 1, 0, 1],                                    [0, 1]
  [0, 0, 0, 0],                                  ]
  [1, 0, 1, 0],
  [0, 1, 0, 1]
]
```

Reconstruction program 1:

```
def partition():
  return [
    {"top left": (0,0), "reflect": False, "rotations": 0},
    {"top left": (0,2), "reflect": False, "rotations": 0},
    {"top left": (3,0), "reflect": False, "rotations": 0},
    {"top left": (3,2), "reflect": False, "rotations": 0},
  ]
```

Reconstruction program 2:

```
def partition():
  copies = []
  for tl_i, tl_j in [(0,0),(0,2),(3,0),(3,2)]:
    copies.append(
      {"top left": (tl_i, tl_j), "reflect": False, "rotations": 0},
    )
  return copies
```

**Example 2**

Input map                                          Unit

```
[                                                  [
  [1,1,1,1,1,1],                                     [1,1,1],
  [0,0,1,0,0,1],                                     [0,0,1],
  [0,0,1,0,0,1],                                     [0,0,1],
  [1,0,1,1,0,1],                                     [1,0,1]
  [0,0,0,0,0,0],                                   ]
  [1,1,1,1,1,1],
  [1,0,1,1,0,1],
  [0,0,1,0,0,1],
  [0,0,1,0,0,1]
]
```

Reconstruction program 1:

```
def partition():
    return [
        {"top left": (0,0), "reflect": False, "rotations": 0},
        {"top left": (0,3), "reflect": False, "rotations": 0},
        {"top left": (5,0), "reflect": True, "rotations": 2},
        {"top left": (5,3), "reflect": True, "rotations": 2},
    ]
```

Reconstruction program 2:

```
def partition():
    copies = []
    for tl_i, tl_j in [(0,0),(0,3)]:
        copies.append(
            {"top left": (tl_i, tl_j), "reflect": False, "rotation": 0},
        )
    for tl_i, tl_j in [(5,0),(5,3)]:
        copies.append(
            {"top left": (tl_i, tl_j), "reflect": True, "rotations": 2},
        )
    return corners
```

**Example 3**

Input map                                          Unit

```
[                                                  [
  [1,1,1,1,1,1],                                     [1,1],
  [1,0,1,0,1,0],                                     [1,0],
  [1,0,1,0,1,0],                                     [1,0]
  [0,0,0,0,0,0],                                   ]
  [0,1,0,1,0,1],
  [0,1,0,1,0,1],
  [1,1,1,1,1,1]
]
```

# H PSEUDOCODE

---

**Algorithm 1** Single-prompt Generative Map Module

---

**Require:** $I$ : Input map, $\quad t$ : Threshold, $\quad C$ : Number of completions, $\quad S$ : Likelihood function

**Ensure:** $\lambda$ : Generative program, $\quad u$ : Unit

1: $S' \leftarrow 0$
2: $\lambda \leftarrow$ ""
3: $u \leftarrow$ ""
4: **while** $S' < t$ **do**
5: $\quad$ Generate a prompt from $I$
6: $\quad$ Send the prompt and receive $C$ completions $(\lambda_1, u_1), \ldots, (\lambda_C, u_C)$
7: $\quad$ Extract Python programs $\{\lambda_1, \lambda_2, \ldots, \lambda_C\}$
8: $\quad$ **for all** $\lambda_i \in \{\lambda_1, \ldots, \lambda_C\}$ **do**
9: $\quad\quad$ **if** $\lambda_i$ runs successfully **then**
10: $\quad\quad\quad$ $S_i \leftarrow S(\lambda_i)$
11: $\quad\quad$ **end if**
12: $\quad$ **end for**
13: $\quad$ $(S', i) \leftarrow \max_i S_i$ $\quad\quad\quad\quad\quad\quad\quad\quad\quad$ ▷ highest scoring program based on likelihood
14: $\quad$ $\lambda \leftarrow \lambda_i$
15: $\quad$ $u \leftarrow u_i$
16: **end while**
17: **return** $\lambda$, $u$

---

**Algorithm 2** Multi-prompt Generative Map Module

---

**Require:** $I_c$ : Input map, $\quad I_p$ : Partial map, $\quad t$ : Threshold, $\quad C$ : Number of completions,
$\quad$ $S$ : Likelihood function

**Ensure:** $\lambda$ : Generative program, $\quad u$ : unit

1: $S' \leftarrow 0$
2: $\lambda \leftarrow$ ""
3: $u \leftarrow$ ""
4: **while** $S' < t$ **do**
5: $\quad$ Generate a prompt from $I_p$
6: $\quad$ Send the prompt and receive $C$ unit candidates $u_1, \ldots, u_C$
7: $\quad$ **for all** $u_i \in \{u_1, \ldots, u_C\}$ **do**
8: $\quad\quad$ Generate a prompt from $I_c$ and $u_i$
9: $\quad\quad$ Send the prompt and receive program $\lambda_i$
10: $\quad\quad$ **if** $\lambda_i$ runs successfully **then**
11: $\quad\quad\quad$ $S_i \leftarrow S(\lambda_i)$
12: $\quad\quad$ **end if**
13: $\quad$ **end for**
14: $\quad$ $(S', i) \leftarrow \max_i S_i$ $\quad\quad\quad\quad\quad\quad\quad\quad\quad$ ▷ highest scoring program based on likelihood
15: $\quad$ $\lambda \leftarrow \lambda_i$
16: $\quad$ $u \leftarrow u_i$
17: **end while**
18: **return** $\lambda$, $u$

---

---

**Algorithm 3** Structure-Based Planner

---

**Require:** $I$ : Input map,    $u$ : Unit,    $C_i$ : Unit copies
**Ensure:** $P$ : Agent path
 1: $(r, c) \leftarrow$ Initial agent position
 2: $C \leftarrow$ Empty set to track fully explored unit copies
 3: $\pi \leftarrow$ Policy for unit exploration based on location of entrance
 4: **while** $C$ does not contain all copies **do**
 5:     Run POMCP on $I$ until reaching an unexplored unit $C_i$

 6:     Identify the current entrance $e$ into $C_i$
 7:     **if** policy from $e$ found in $\pi$ **then**
 8:         Explore $C_i$ with policy from $\pi$
 9:         Add $C_i$ to $C$
10:     **else**
11:         Run POMCP on $C_i$ explore the unit
12:         Add new policy $(e, \pi_e)$ to $\pi$
13:         Add $C_i$ to $C$
14:     **end if**

15:     $o_1, \ldots, o_m \leftarrow$ Exit locations of $C_i$
16:     **for** $o_j$ in $\{o_1, \ldots, o_m\}$ **do**
17:         $p_j \leftarrow$ compute penalty for escaping $C_i$ from $o_j$
18:     **end for**
19:     Run MCTS on $C_i$ until reaching an exit to escape
20: **end while**

21: Run POMCP on $I$ to explore the rest of the map
22: **return** $P$

---

**Algorithm 4** POMCP for In-Unit Planning

---

 1: **procedure** SEARCH($h$)
 2:     **if** $B(h) = \emptyset$ **then**
 3:         **return**
 4:     **else**
 5:         **repeat**
 6:             $s_{\text{exit}} \sim B(h)$
 7:             Simulate($s_{\text{exit}}, h, 0$)
 8:         **until** Timeout
 9:         **return** $\arg\max_a V(ha)$
10:     **end if**
11: **end procedure**
12:
13: **procedure** ROLLOUT($s_{\text{exit}}, h, \text{depth}$)
14:     **if** depth $>$ depth limit **then**
15:         **return** 0
16:     **end if**
17:     $a \sim \pi_{\text{random}}$
18:     $(o, r) \sim \mathcal{G}(h, a)$
19:     **if** $o$ contains $s_{\text{exit}}$ **then**
20:         **return** $r$
21:     **else**
22:         **return** $r + $Rollout($s_{\text{exit}}, ha, \text{depth}+1$)
23:     **end if**
24: **end procedure**

 1: **procedure** SIMULATE($s_{\text{exit}}, h, \text{depth}$)
 2:     $N(h) \leftarrow N(h) + 1$
 3:     **if** depth $>$ depth limit **then**
 4:         **return** 0
 5:     **end if**
 6:     **if** $h \notin T$ **then**
 7:         **for** $a \in \{\text{up}, \text{right}, \text{bottom}, \text{left}\}$ **do**
 8:             $T(ha) \leftarrow (N_{\text{init}}(ha), V_{\text{init}}(ha), \emptyset)$
 9:         **end for**
10:         **return** Rollout($s_{\text{exit}}, h, \text{depth}$)
11:     **end if**
12:     $a \leftarrow \arg\max_a \left[ V(ha) + c\sqrt{\frac{\log N(h)}{N(ha)}} \right]$
13:     $(o, r) \sim \mathcal{G}(h, a)$
14:     **if** $o$ contains $s_{\text{exit}}$ **then**
15:         $N(ha) \leftarrow N(ha) + 1$
16:     **else**
17:         $r \leftarrow r + $Simulate($s_{\text{exit}}, ha, \text{depth}+1$)
18:     **end if**
19:     $V(ha) \leftarrow V(ha) + \frac{r - V(ha)}{N(ha)}$
20:     **return** $r$
21: **end procedure**

---

# I   HUMAN EXPERIMENT - MAZE SEARCH TASK

This study runs best on a desktop/laptop.

The study will **NOT** run on Safari, or a mobile device.

In this study you will look for an exit in a maze.

After this, you will be asked to provide demographic information.

The study is expected to take about 10 minutes.

Thanks for participating!

Figure 10: Introductory screen.

**INSTRUCTIONS (PLEASE READ CAREFULLY)**

Your task is to exit the maze by reaching the red square, which is initially hidden.

You can move one square at a time by clicking on the white squares next to your character.

You cannot see through the walls. The squares you cannot see yet are black.

The exit could be behind any of the black squares.

A maze looks like this:

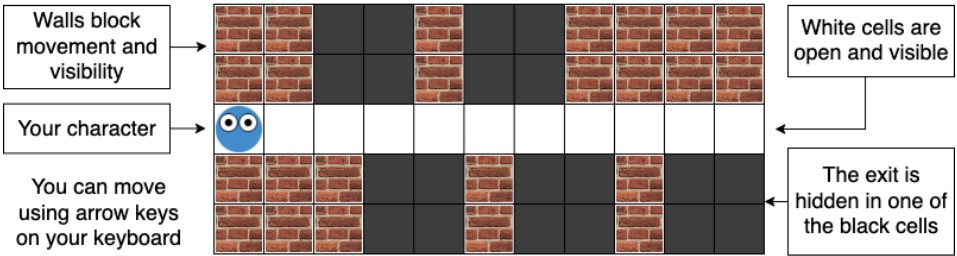

Figure 11: Instructions.

**Practice Maze 1 of 5**

Let's look at this map. There are some black squares, a brick wall, and your character.

The exit could be behind any of the black cells.

You can move your character by clicking one of adjacent white cells.

Find the exit and step on it to exit the maze.

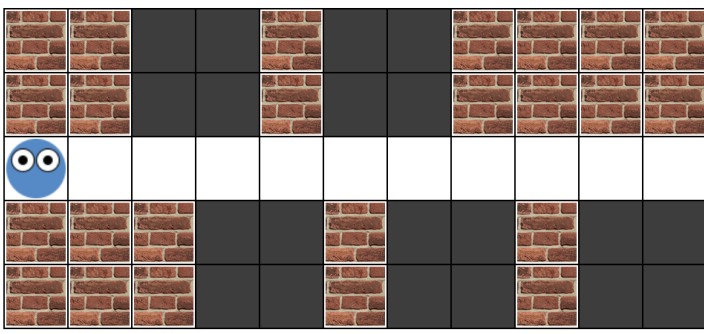

Figure 12: Practice (there are 5 practice trials).

Please answer the quiz to move on.

## Question 1: My task is to ..

○ visit every square in the maze
○ there is no specific task
○ find the exit in each maze
○ click as fast as possible

## Question 2: Exits are always placed ...

○ in the bottom left corner
○ in one of the black cells
○ some mazes have no exit
○ there may be multiple exits

## Question 3: Which image correctly shows unseen parts of the maze?

○ Image A    ○ Image B

A.                                    B.

Figure 13: Comprehension Quiz.

**Maze 1 of 21**

Find the exit and step on it to exit the maze.

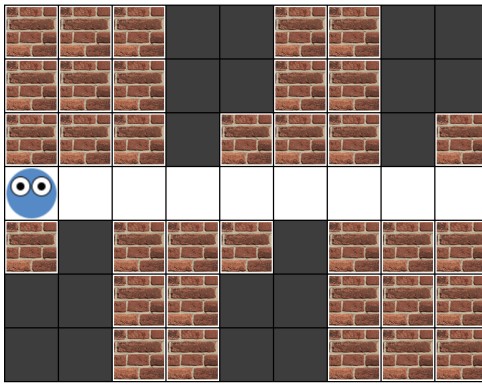

Figure 14: Experiment view.

