# OpenReview forum: "Planning with Generative Cognitive Maps"
_ICLR.cc/2026/Conference — Submitted to ICLR 2026_

### Official Review · Reviewer_rQQ4 · 2025-10-27

**Soundness:** 2
**Presentation:** 3
**Contribution:** 1
**Rating:** 2
**Confidence:** 4

**Summary:**

This paper studies structured planning, with a focus on the synthesis of compressed cognitive maps that retain only the most planning-salient information about an environment. The work's goal is twofold: first, to develop an explanatory algorithmic account of how people might be forming simplified cognitive maps and planning with them, and second, to advocate for this approach as a generic algorithmic template for structure based planning in practice, inspired by how people plan. Maze Search Task is used as the central planning domain, a simple partially observable grid game that has been of recent interest to similar work in which the player must exit the maze. The core of the approach is a two-piece algorithmic recipe: program induction over possible cognitive maps is used to hand off a transition function to a structured planner. Together, these pieces constitute "Gen-POMCP", a generative planner. This approach is contrasted with standard POMCP (without the program induction over maps), and a variant of POMCP that incorporates a cognitive constraint (Depth Plan). A human study is then conducted to draw out the explanatory power of these different hypotheses. Behavioral metrics are deployed to compare DepthPlan with Gen-POMCP in the user study. The primary finding is that Gen-POMCP is a better predictor of human behavior than DepthPlan.

**Strengths:**

- The question to better understand how people abstract and plan is a deep and important one. The ingredients in the proposed theory are intuitive, and resonate with other similar methods in the literature.
- The paper was easy to read, and the core of the method and experimental design are intuitive (with a minor exception regarding the details of the method, I detail this below).
- Inclusion of a human study, and the initial results contrasting Gen-POMCP with DepthPlan, are compelling.

**Weaknesses:**

W1. I am broadly sympathetic to the research agenda to understand how people plan. However, one stated contribution of the work is "...an algorithmic account of how structure-based planning can be implemented in practice". There are large bodies of literature within the ICAPS, AI, and RL communities dedicated to this question. As such, I believe the present work, to be treated as a novel algorithmic account of structure-based planning, needs to be contrasted using the same language and analysis that are used to evaluate planning algorithms (or, a convincing argument should be made about why different analysis tools are needed). For example, recent work by Wen et al. explicitly proves under what conditions structured planning can be efficient (Propositions 1-3). Or, early work on planning explores how structure changes the planning problem, as in ABSTRIPs by Sacerdoti (1974). Or, more recently, Jinnai et al. (2018) examine the computational hardness of finding structures that can make planning as easy as possible, and prove both that this problem is NP-Hard in general, but that efficient approximation algorithms can exist (along with an average-case result follow-up by Ivanov et al.). In this sense, there are many known results for understanding when and why we can design efficient algorithms for structure-based planning. I struggle to see the novelty here from an AI perspective, especially since (1) the basic mechanisms used to scrutinize planning algorithms are not engaged with (computational hardness, robustness, and so on), and (2) the work is not situated conceptually or experimentally relative to the vast body of literature that studies structure-based planning algorithms. As further examples, consider work by Anand et al. (2016) that develop "on the go" abstractions to facilitate tree-based search, or Hostetler et al. (2014), who make use of state aggregation to simplify MCTS, or seminal work by Littman (1997) that proves various hardness results about different kinds of planning. Or, we can look at hierarchical structure, as in SHOP (Nau, 1999) or HTNs (Erol, 1997).

W2. Much of the terminology and notation are under-developed or lacking in rigor and depth. For instance, planning is described as search through a tree---this is one plausible account of planning, though is still only part of the story. See, for instance, the work by Littman, who gives precise formal definitions to certain variants of planning problems. Similarly, POMDPs are defined using a standard definition of Belief MDPs, which is non-standard; the introduced object is an MDP over belief state, rather than a POMDP (with Kaelbling et al. showing how to connect the two). See the standard POMDP definition in the cited Kaelbling et al. paper in Section 3.1. I can imagine the Belief MDP definition is introduced for a reason, but it would be more effective to introduce POMDPs in full detail, then describe their translation into a Markov process over belief state.

W3. The primary contribution of the work is to offer an explanatory algorithmic account about how people might be carrying out structured planning in POMDPs. The evidence offered to support the conclusion that Gen-POMCP is a valid such explanatory account is relatively thin; a variety of competing accounts are described in Section 4, but the model predictions from these alternative theories are not contrasted with. For instance, the work by Ho et al. is described as showing "that people form cognitive maps that facilitate planning ... by selectively representing only the goal-relevant parts of the map". This sounds extremely similar to the proposal in this paper: I did not understand where the two theories come apart, and where one might have higher explanatory power than the other.

W4. Lastly, the core method is not described in adequate detail. Pseudo-code will add clarity to what precisely is being proposed by Gen-POMCP.

To summarise: I perceive the stated contributed as spread across both (1) developing new structure-based planning algorithms inspired by compositional accounts of how people represent cognitive maps, and (2) a proposed explanatory account of how people form cognitive maps and plan with them. The former is missing contact with the rigor, literature, and core analysis tools that evaluate approaches to (structure-based) planning, and the latter is missing direct comparison to other models of how people form simplified representations and plan. In light of this, I take the core claims to not yet be well-supported by evidence, and so I recommend rejection at this time. I am happy to discuss during rebuttal if I have misunderstood, or if there are other factors that can remedy these issues.

Typos and Writing Suggestions:

- " Solving there problems optimally " --> maybe,  "Solving these problems optimally "?
- "In the the remainder of this section..."

References:
- Anand, A., Noothigattu, R., & Singla, P. (2016, March). Oga-uct: On-the-go abstractions in uct. In Proceedings of the International Conference on Automated Planning and Scheduling (Vol. 26, pp. 29-37).
- Hostetler, J., Fern, A., & Dietterich, T. (2014, June). State aggregation in Monte Carlo tree search. In Proceedings of the AAAI Conference on Artificial Intelligence (Vol. 28, No. 1).
- Littman, M. L. (1997). Probabilistic propositional planning: Representations and complexity. AAAI/IAAI, 748-754.
- Ivanov, A., Bagaria, A., & Konidaris, G. (2025, April). Discovering Options That Minimize Average Planning Time. In Proceedings of the AAAI Conference on Artificial Intelligence (Vol. 39, No. 17, pp. 17573-17581).
- Jinnai, Y., Abel, D., Hershkowitz, D., Littman, M., & Konidaris, G. (2019, May). Finding options that minimize planning time. In International Conference on Machine Learning (pp. 3120-3129). PMLR.
- Nau, D., Cao, Y., Lotem, A., & Munoz-Avila, H. (1999, July). SHOP: Simple hierarchical ordered planner. In Proceedings of the 16th international joint conference on Artificial intelligence-Volume 2 (pp. 968-973).
- Erol, K. (1995). Hierarchical task network planning: formalization, analysis, and implementation. University of Maryland, College Park.
- Sacerdoti, E. D. (1974). Planning in a hierarchy of abstraction spaces. Artificial intelligence, 5(2), 115-135.
- Wen, Z., Precup, D., Ibrahimi, M., Barreto, A., Van Roy, B., & Singh, S. (2020). On efficiency in hierarchical reinforcement learning. Advances in Neural Information Processing Systems, 33, 6708-6718.

**Questions:**

Q1: What are the main criteria by which we should be evaluating a planning algorithm, and how does Gen-POMCP fair relative to these criteria? How does it situate relative to other algorithms---for instance, in the planning community, to things like Hierarchical Task Networks (HTNs) or SHOP or ABSTRIPS? Or, alternatively, is there a reason we should not be contrasting the present approach to other planning algorithms in this way?

Q2: The basic mechanism described by line 281 sounds similar to prior methods for decomposing planning into subproblems (as in HTNs). Can you spell out in more detail what is happening, and comment on how it might differ from standard hierarchical planning?

And, one minor question:

Q1. It is unclear what role partial observability plays here. Planning in a POMDP vs. an MDP are quite different---how central is partial observability to the approach?

---

> ### Author Response · Authors · 2025-11-24
> **Response to Reviewer 4**
>
> We thank the Reviewer for constructive comments, typos, and references. Below we respond to the Reviewer's comments point by point.
>
> __Q1__
>
> Thank you for the helpful pointers to related work in hierarchical planning, HRL, and option discovery. We will incorporate them in the revised manuscript.
> Our goal in this paper is to give a computational account of how people plan in structured environments by integrating (i) generative map reconstruction and (ii) modular planning within that recovered structure. The choice of POMCP for planning is based on prior ICLR paper on human planning ( Sharma et al. 2022). As our aim is to explain behaviour, we do not provide a full theoretical evaluation of Gen-POMCP (e.g., search efficiency, completeness, admissibility). HTNs, SHOP, ABSTRIPS, and related systems are engineered for those criteria; our model is instead designed to test whether inductive biases and modular decompositions predict human choices.
>
> That said, in Appendix B we theoretically analyze SBP by providing worst-case bounds on step-cost differences between Gen-POMCP and Naive-POMCP -- showing that SBP incurs a constant-factor step penalty, which does not change the asymptotic exploration cost. The cost difference between modular and global planners is only constant-factor, and in non-fragmented settings the worst-case costs coincide exactly. We hope our work will motivate further theoretical research on theoretical bounds of structure-based planning.
>
> __Q2__
>
> The procedure introduced in the SBP is not an HTN-style decomposition. The SBP describes structure-conditioned policy reuse induced by a generative map representation, not a hierarchical task decomposition in the classical HRL sense.
>
> - The GMM clusters the maze into repeated structural fragments.
> - SBP treats each fragment as an isomorphic subproblem, solving a local POMCP within each fragment and reusing that solution whenever the same fragment is instantiated.
> - At the “top level,” SBP chooses which fragment to enter next.
>
> How this differs from HTNs, SHOP, ABSTRIPS:
>
> - No hand-engineered task hierarchy. SBP Fragment boundaries come purely from the learned generative structure of the map.
> - GMM induces a state-space compression, not action abstraction. SBP’s hierarchy arises from regularities in this state-space (repeated fragments) rather than abstractions over actions or goals. It is a representation-level decomposition, not task-level.
> - We focus on policy reuse, not refinement. An HTN refines tasks recursively, wheres SBP reuses local POMCP policies across isomorphic fragments.
> - The “modules” exist only because the GMM inferred them; they may not align with the causal/action structure that HTNs rely on.
> - SBP  is not presented as an improved hierarchical planner over algorithms like HTN, but as an illustration of how a planner can work with compositional structure discovered by the GMM given cognitive compute constraints.
>
> __Q3__
>
> Partial observability in the MazeSearch environment lies in occluding parts of a maze, so that the agent must maintain beliefs over unobserved regions. In MazeSearch Task people can always see the walls, and we provide GenPlan with the same information. The observed area can be reduced without the loss of generality, as long as GenPlan observes more than one instantiation of a given structural unit.
>
> __Comparison to prior work on human planning__
>
> Similar to our work, Tthe work by Ho et al. shows that cognitive maps of grid-worlds are not exact representations, however this work does not deal with regular structure. Rather, it tests how people recall unstructured maps they have previously navigated, which does not allow for a direct comparison.
>
> Currently,  (Sharma et. al. 2022) is the only study that addresses human planing in structured domains. However, their task domain demands that the map is perfectly tiled by a given fragment (including its reflections, as specified by a given CFG).  They show that people use prior expectations of structure to predict maps, however their experiment can not differentiate between Gen-POMCP and Naive-POMCP as both make identical predictions. In contrast, we show that beyond anticipating maps, map structure itself shapes how people make plans in such environments.
>
> __Novelty__
>
> We introduce a novel sample-efficient program induction over programs in a Turing-complete language to model how structure is inferred from observations. Our method improves on a prior ICLR paper (Sharma et. al. 2022), which models map inference in structured environments via exhaustive enumeration over structures specified by a rigid a context-free grammar.
> We offer an algorithmic account of how people may plan under uncertainty in structured environments, if given strong priors about the world and limited cognitive resources.
>
> _We will clarify our contribution in an upcoming revision, provide pseudocode for our GMM and SBP framework, and expand discussion of related works, to be uploaded by Dec 1_

---

### Official Review · Reviewer_sVXi · 2025-10-28

**Soundness:** 3
**Presentation:** 3
**Contribution:** 4
**Rating:** 6
**Confidence:** 3

**Summary:**

In this paper, Planning with Generative Cognitive Maps, the authors designed GenPlan, a computational framework that represents a task environment as a generative cognitive map and plans by reusing policies on repeating structures, to model human navigation behavior. GenPlan consists of a Generative Map Module (GMM) that infers compositional generative representation of the environment structure through a programmatic map generated by LLM (GPT4), and a Structure-Based Planner (SBP) that leverages the learned structure to perform navigation planning with policy reuse.

Empirically, GenPlan reproduces planning behavior more similar to that of human participants compared with naive POMCP (giving rise to a global optimal policy) and constrained POMCP (with discounts to limit planning depth). Moreover, GenPlan significantly reduces computational planning cost: in simulations on larger structured mazes, it required far fewer rollouts to fully explore the environment than a standard planner, while still finding the goal with only a modest increase in path length. Overall, the paper’s contributions include a novel generative model of cognitive maps (GenPlan), an integrated planning algorithm (Gen-POMCP) that leverages that model, and experimental evidence suggesting that humans similarly exploit compositional structure to plan efficiently.

**Strengths:**

1. The work introduces a novel framework by uniting generative modeling and planning in the context of cognitive maps, and provides scientifically significant results for cognitive science. What's particularly novel about the approach is the use of a Large Language Model (LLM) to induce a programmatic map for environment representation. This allows tractable inference of structure where previous methods would require potentially intractable enumeration. The framework (GenPlan) is thus novel in how it integrates an LLM-induced programmatic structural prior with a planning algorithm – a creative synthesis of ideas from program induction, hierarchical planning, and cognitive psychology. The cognitive experiment shows a higher human similarity for GenPlan model, indicating that human deviations from optimal policies are, at least in part, due to approximating planning by piecewise policies conditioned on structure for policy reuse.
2. The paper is technically solid and provides validation on its base claims. The GenPlan framework is described with a clear formal basis, with the planning problem as a POMDP, and the model is compared against relevant baseline cognitive models, including DepthPlan and naive POMCP. The experiments are relatively well designed for model and human subjects to test the behavioral hypotheses and evaluate performance.

**Weaknesses:**

1. While environments of many different shapes are used for the experiments, all the environments are compositionally simple, consisting of the same top-level grid maze structure with identical sub-mazes per environment. This limits the generalizability of the hypothesis due to the limited scope of the environmental composition (2-level maze with identical substructure).
2. GenPlan's GMM steps involves using GPT4 over the entire environment for producing a programmatic map generator. This requires global knowledge about the environment as prior knowledge. In the experimental description, it wasn't clear that the human subjects were provided with the similar prior knowledge. This might hinder the validity of the comparison.

**Questions:**

1. Could the authors clarify how exactly the LLM is used to infer the generative map structure? How is the LLM prompted?
2. What prior knowledge do the human subjects have for the experiments? For example, are they informed that the maze has a similar substructure?
3. There are limited performance evaluation and ablation studies for evaluating GenPlan comprehensively. How do the parameters of GenPlan affect the results?

---

> ### Author Response · Authors · 2025-11-16
> **Response to Reviewer 3**
>
> We thank the Reviewer for the constructive comments and questions. Below we respond to the Reviewer's comments point by point.
>
> __1. How exactly the LLM is used and prompted?__
>
> *We will include full prompts and  GMM pseudocode in the revised SI by Dec 1.
>
> Below we provide a concise description of our process.
>
> LLM is given a map encoded as a numpy binary array and is asked to generate  (1) a candidate structural unit from which this map can be restored, and
> (2) a Python program that reconstructs the map from that fragment. We provide a starter Python code in the prompt, and two examples of input --> output.
>
> Example response:
>
> ```
> import numpy as np
>
>
>
> # input
>
> input_map =[
>
> [1,1,1,0,0,1,1,0,0],
>
> [1,1,1,0,0,1,1,0,0],
>
> [1,1,1,0,1,1,1,0,1],
>
> [0,0,0,0,0,0,0,0,0],
>
> [1,0,1,1,1,0,1,1,1],
>
> [0,0,1,1,0,0,1,1,1],
>
> [0,0,1,1,0,0,1,1,1]
>
> ]
>
>
>
> # fragment
>
> fragment = [
>
> [1,1,0,0],
>
> [1,1,0,0],
>
> [1,1,0,1]
>
> ]
>
>
>
>
> def construct_copy(fragment, reflect, rotations):
>
> if reflect:
>
> fragment = np.flip(fragment, 1)
>
> fragment = np.rot90(fragment, rotations)
>
> return fragment
>
>
>
> def regenerate_pattern(fragment, copies, input_dims):
>
> pattern = 0.5 * np.ones(shape=input_dims)
>
> for copy in copies:
>
> transformed = construct_copy(
>
> fragment,
>
> copy["reflect"],
>
> copy["rotations"],
>
> )
>
> height, width = transformed.shape
>
> tl_i, tl_j = copy["top left"]
>
> try:
>
> pattern[tl_i : tl_i+height, tl_j : tl_j+width] = transformed
>
> except:
>
> pass
>
> return pattern
>
>
>
> def partition():
>
> copies = [
>
> {"top left": (0,1), "reflect": False, "rotations": 0},
>
> {"top left": (0,5), "reflect": False, "rotations": 0},
>
> {"top left": (4,0), "reflect": False, "rotations": 2},
>
> {"top left": (4,4), "reflect": False, "rotations": 2}
>
> ]
>
> return copies
>
>
>
> output = regenerate_pattern(fragment, partition(), np.array(input_map).shape)
> ```
>
> We solicit a small number of prompt completions (5-10) which are scored based on reconstruction quality to approximation a distribution over cognitive maps. The exact wording of the prompt is not critical to GenPlan architecture; Effectively, as long as we obtain at least one reasonable map reconstruction from GMM, such that correctly identifies at least one structural unit and at least one recurrence of this unit, GenPlan is able to save planning compute costs.
>
> __2. What prior knowledge do the human subjects have for the experiments?___
>
> *We will include full Experiment Instructions in the revised SI by Dec 1. *
>
> People are only told that they need to find an exit from a maze, which can be anywhere in the maze, suggesting a uniform prior distribution. They are not told that the maps are structured in a particular way, however in the MazeSearch setup the wall locations can be seen from the outset, as people can see the grid-world maze from an overhed view. We encourage the reviewer to try the task here:
> http://18.25.132.241/fragments/int_exp.php
>
> __3. Limited ablation studies. How do the parameters of GenPlan affect the results?__
>
> Free parameters are weights for reconstruction accuracy and planning costs.
>
> *Reconstruction accuracy.*
> When map reconstruction accuracy decreases, the amount of online heuristic planning increases, and the amount of (structure-based) planning decreases. Our implementation uses the best fitting myopic heuristic documented in prior work with Maze Search (Kryven et al 2024), that goes toward the nearest unobserved area (similar to a BestFirst algorithm).
>
> *Planning cost.*
> When planning cost increases, the fragments become smaller - leading the model to produce a more localized search. A negligible planning reduces the model to a global POMCP planner.
>
> Parameter combinations could produce different stereotyped behaviours:
>
> High reconstruction accuracy, low planning cost -- optimal POMCP.
>
> Low reconstruction accuracy, any planning cost -- heuristic.
>
> High reconstruction accuracy, high planning cost: -- increasingly modular with (increasing modularity with increasing planning cost).
>
> Since the goal of the current study is to test whether people use structure-based as opposed to global search, our experiment does not dissociate between different parametrization of structure-based planning. We fit a population-level model that keep planning cost high, so it can plan within units built-in by design to maximize the amount of discriminating decisions between structure-based and global planning. The effect of using the MLE map is to maximize map reconstruction accuracy given constraints on the size of the structural unit.

---

### Official Review · Reviewer_mVRx · 2025-10-29

**Soundness:** 2
**Presentation:** 2
**Contribution:** 2
**Rating:** 2
**Confidence:** 4

**Summary:**

The paper proposes GenPlan, a cognitively grounded framework for planning in structured, partially observable environments. It has two parts: (i) a Generative Map Module (GMM) that infers a small set of repeated “structural units” and a program that reconstructs the map using transformations (rotations/reflections), guided by a likelihood balancing reconstruction accuracy, description length (MDL), and unit informativeness; and (ii) a Structure-Based Planner (SBP) that plans within each inferred unit (via POMCP/MCTS) and between units using a heuristic that prefers unit-boundary cells with shorter average Manhattan distance to remaining unseen cells.
GenPlan’s predictions align with human action choices better than a depth-limited global planner; computationally, GenPlan searches large, structured mazes with far fewer rollouts than a structure-naïve POMCP while producing good-enough paths.

**Strengths:**

1. Inspired by resource rationality and compositionality in cognitive science, the work proposes generative, program-based representation of cognitive maps and uses it directly for policy reuse which can both explain human behavior and produce efficient plan.
2. The paper is clearly written and structured.

**Weaknesses:**

1. Though human study is provided and considered, the experiment maps are relatively toy.
    - This is actually a simplified POMDP environment. 1) Walls are assumed known from the start; only the exit is hidden, which removes core POMDP uncertainty about topology and sensing. 2) How to handle non-deterministic maps?
    - Although a posterior over generative maps is defined, the implementation selects only the MAP reconstruction for planning, avoiding online uncertainty resolution. How does it work if the model has to plan over a distribution of maps?
    - All behavioral mazes are structured with 2–20 repeating units covering 80-100% of the layout, which is exactly the condition GenPlan is built to exploit. It would be much better if you consider working on more realistic scenarios, or even real-world maps.
2. It seems like the hierarchial RL and options are completely omitted, both in discussion the connection with GenPlan, and baseline models.

**Questions:**

See weakness.

---

> ### Author Response · Authors · 2025-11-14
> **Response to Reviewer 2**
>
> We thank the Reviewer for constructive discussion. Below we respond to the Reviewer's comments point by point.
>
> 1. Toy environment used.
>
> We'd like to clarify that since our goal is to understand cognitive computations in structured spatial planning, our choice of  environment is motivated by prior SOTA models of human spatial planning, which used the MazeSearch environment. This choice of environment enables us to directly compare our model to DepthPlan.
> MazeSearch accommodates uncertainty by assuming that the agent sees only a part of the map. By default, the amount of information given to GMM is the same as given to humans in the MazeSearch paradigm, that is, and includes locations of all walls (rewards are hidden). However, without the loss of generality, GMM can recover structural units from any observed part of the map, regardless if larger parts of the map are occluded.
>
> 2. How to handle non-deterministic maps?
>
> We currently assume that map structure is deterministic, meaning that the units are stable within the environment. This setup aligns with prior works on human spatial planning, which all use deterministic maps (Ho at al, 2022; Kryven et al, 2024; Sharma at al 2022), allowing us to compare our model to prior work.
>
> To work with probabilistic units, GenPLan would need to use a different implementation of GMM and SBP , although the same principles of local structure-based planning will still apply.
> For GMM to encode probabilistic units, the unit maps will need to be specified by generative programs using a graph-grammar or a probabilistic CFGc based on symbolic primitives - e.g. corridors, different types of junctions. One could still use LLM as a model of prior knowledge over what these primitives may be. Such probabilistic unit implementation is currently outside the scope of our work.
>
> 3. How does it work if the model has to plan over a distribution of maps?
>
> Reasoning over a distribution over maps is easy to implement, however our experiment design can not distinguish between MLE and distribution-based reasoning. We elaborate below.
>
> When GMM proposes a sample of maps, the probability distribution over these maps is proportional to their score. SBP can  compute a probability distribution over actions in either one map, or across or all of these maps, and marginalize actions them:  P(action_j|) =\sum_i P(action_j|map_i)P(map_i).
>
> However, as shown in prior work (Sharma at al 2022), finding test-cases where the use of MLE and distribution over maps produce different predictions can be hard. For instance, these cases occur when using a latent cue, where the user's behaviour of searching for the cue before searching for reward is diagnostic of distribution-based reasoning. To do this, (Sharma at al 2022) used hidden markers that the user had to discover as a shortcut to subsequent reward placement.
> The MazeSearch domain does not have the capacity to place such markers. Therefore, MLE map is a reasonable approximation for our proof-of-concept implementation.
>
> *We will clarify this point in a revision, to be uploaded by Dec 1.*
>
> 4. It would be much better if you consider working on more realistic scenarios, or even real-world maps.
>
> We find the question of real-world generalization very interesting and intend to explore it in future work, as a follow-up to demonstrating structure-based planning in human behaviour.
>
> *We will revise the Discussion to discuss to this point by Dec 1*
>
> 5. It seems like the hierarchial RL and options are completely omitted.
>
> *We've made references to HRL the Related Work section, and will expand this section to discuss more HRL literature by Dec 1*

---

### Official Review · Reviewer_kTcK · 2025-11-10

**Soundness:** 2
**Presentation:** 2
**Contribution:** 2
**Rating:** 2
**Confidence:** 4

**Summary:**

This paper introduces GenPlan, a computational framework that explains how humans plan efficiently in structured environments by using generative cognitive maps that capture compositional structure through repeated patterns and symmetries. The framework consists of two key components: a Generative Map Module (GMM) that discovers programmatic representations of environments using LLM-based program synthesis, and a Structure-Based Planner (SBP) that reduces planning costs by reusing policies across structural units rather than computing global optimal solutions. The authors validate their approach through a behavioral experiment with 30 participants in a Maze Search Task, demonstrating that human planning aligns significantly better with the structure-exploiting Gen-POMCP model than with the previous state-of-the-art depth-limited planning model, and they provide simulation evidence that their approach achieves substantially lower computational cost than naive planning while maintaining good solution quality.

**Strengths:**

The paper makes both a scientific contribution by showing that human deviations from optimal policies reflect structure-based planning with policy reuse, and an engineering contribution by demonstrating how to implement these principles in a working system. Across all environments, Gen-POMCP predicts human behavior significantly better than DepthPlan, the previous state-of-the-art model, with people showing high consistency with structure-based planning across all environments and individuals.

**Weaknesses:**

Foremost, experiment recruited 30 participants from Prolific in a single maze search domain that, despite being structured, remains relatively artificial compared to real-world planning contexts. The framework makes simplifying assumptions, including using the most likely generative map rather than maintaining a distribution over maps, and assuming stable population-level weights for reconstruction accuracy and planning costs.
Also the approach's heavy reliance on LLM-based program synthesis introduces potential biases: the structural discovery depends on how the LLM is prompted and the particular priors it has learned, yet there is limited analysis of robustness to these design choices.
Lastly, actually there are line of concurrent works trying to do offline planning with the power of LLMs. They are similar to your work trying to utilize the power of LLM to plan before action, hope you can cite them and highlight your novelty.
Shinn et al. Reflexion: Language Agents with Verbal Reinforcement Learning
Zelikman et al. Parsel: Algorithmic Reasoning with Language Models by Composing Decompositions
Kim et al. How language models extrapolate outside the training data: A case study in Textualized Gridworld
Yang et al. Chain of Thought Imitation with Procedure Cloning
Yao et al. ReAct: Synergizing Reasoning and Acting in Language Models
Gu et al. Is Your LLM Secretly a World Model of the Internet? Model-Based Planning for Web Agents

**Questions:**

How were the free parameters in the likelihood function chosen, and how sensitive are the main results to these choices? The paper notes that planning varies between individuals and proposes that this variability arises from different representations of the same map depending on available cognitive resources, but the implementation assumes fixed population-level weights. How would the framework dynamically adjust these weights to model flexible cognitive resource allocation within individuals? Can the principles extend beyond spatial domains to other planning contexts like recipe planning or multi-step problem solving, and if so, what would constitute the "structural units"? Regarding the LLM-based approach: how much do results depend on using GPT4 specifically, and would different LLMs produce meaningfully different structural decompositions?

---

> ### Author Response · Authors · 2025-11-14
> **Response to Reviewer 1**
>
> We thank the Reviewer for constructive comments and for the suggested citations, which we will incorporate in our revision. Below we respond to the Reviewer's comments point by point.
>
> __1. Behavioural evaluation is limited to N=30 participants, one task domain__
>
> We'd like to clarify our reasons for using this experiment scale. Our choice to use N=30 subject and one task domain is based on a prior ICLR paper on learning cognitive maps (Sharma et. al. 2022 https://arxiv.org/pdf/2110.12301), as well as pilot studies used to power our design, to estimate the number of participants sufficient to demonstrate the effect.
>
> *We will clarify this point in a revision, to be uploaded by Dec 1.*
>
> __2. Using the distribution over generative maps.__
>
> Reasoning over a distribution over maps is easy to implement. In our setup the two methods (MLE and distribution) produce identical behaviour. We elaborate below.
>
> GMM currently proposes a sample of maps. The probability distribution over these maps is proportional to their score (accuracy and MDL).
> To reason over the distribution, SBP needs to compute a probability distribution over actions *for each map*, which is trivial to implement. Then, we can select a single action by marginalizing over maps:  P(action_j|) =\sum_i P(action_j|map_i)P(map_i).
>
> However, as shown in prior work (Sharma at al 2022), finding test-cases where the use of MLE and distribution over maps produce different predictions can be hard. These cases require using a latent cue, where the user's behaviour of searching for the cue before searching for reward is diagnostic of distribution-based reasoning. For instance, (Sharma at al 2022) used a hidden coloured marker that the user had to learn to associate with subsequent reward placement.
> The Maze Search domain, (which we use to compare human behaviour to the SOTA model of human planning), does not have the capacity to place such markers. Therefore, we use MLE map as simplifying assumption that does not affect our proof-of-concept implementation.
>
> *We will clarify this point in a revision, to be uploaded by Dec 1.*
>
> __3. How were the free parameters in the likelihood function chosen, and how sensitive are the main results to these choices?__
>
> Free parameters are reconstruction accuracy and planning costs.
>
> *Reconstruction accuracy.*
> As map accuracy decreases, the amount of online heuristic planning increases, and the amount of structure-based planning decreases. We implement this heuristic based prior work with Maze Search (Kryven et al 2024). A zero reconstruction accuracy entails a fully heuristic planning, regardless of planning cost.
>
> *Planning cost.*
> As planning cost increases the units become smaller, leading to more localized search. A negligible planning cost paired with a high reconstruction accuracy reduces the model to a global planner. A high accuracy and high planning cost leads to a fully structure-based planning (the population level model used in the paper)
>
> In practice, dissociating between these parameters in a human experiment requires a complex targeted design, beyond the scope of the current work.
>
> As our goal is to test whether people use structure-based planning, as opposed to global search in the previous SOTA models, use a population-level model with high reconstruction accuracy and low planning costs. This leads the model to plan within single units intended by design (rather than grouping them) and maximizes the amount of discriminating decisions between structure-based and global planning.
>
> __4. How would the framework dynamically adjust these weights to model flexible cognitive resource allocation within individuals?__
>
> We note two points here
> (1) behaviour differs between individuals;
> (2) behaviour could change within the same individual over time - over successive trials or days
>
> We currently document individual difference as a degree of using modular (vs. global) planning, showing a proof-of-concept of individual differences that underestimates the extent of structure-based planning.
> While out of scope of the current study, an extended experiment could allow for fitting a model that studies both how reconstruction accuracy and planning cost (the scale of the structured unit) change between people and over time.
>
> __5. How much do results depend on using GPT4 specifically?__
>
> We used GPT4 as a baseline for coherent map reconstruction, as most models consistently failed to generate valid candidates from large 2D array input. Any similarly capable model should likely produce the same MLE map despite differences among candidates.
>
> *We will expand on this in the revision, to be uploaded by Dec 1.*
>
> __6. Can the principles extend beyond spatial domains__
>
> We are exploring a generalization of our framework's principles to cognitive graphs - where GMM produces graph-based representations.
>
> *We will clarify this point in a revision, to be uploaded by Dec 1.*

---

### Author Response · Authors · 2025-12-01
**Revision summary**

Dear AC,

To facilitate the AC’s review, we summarize the revisions made to the manuscript below.

The changes to the manuscript are highlighted in blue, and include:
- an expanded our discussion of prior work -- hierarchical RL, and works that use LLM to support planning -- based on references suggested by the reviewers.
- an expanded discussion of limitations clarifying the focus of our work
- New references to a substantially expanded Appendix

The changes to the Appendix include 15 new pages (pages 21 - 36) across Appendices  E, F, G, H, I -- detailing:
- GMMs generalization across four different LLM architectures
- full prompt text
- full algorithm pseudo-code for GMM and SBP
- discussion of generalization to other LLM architectures
- full experimental instructions for human study

Thank you for considering our revision!

The Authors

---

### Meta-Review · Area_Chair_YAph · 2026-01-05

**Summary:**

While reviewers overall agreed that the model was well presented, and that the comparison with human planning was compelling, they raised several important concerns:

__Concerns about the single task environment__
* Three reviewers expressed concerns about the task environment. These included that the experiments were run on "a single maze search domain", that "all the environments are compositionally simple", that the domain "is exactly the condition GenPlan is built to exploit". Instead of addressing any of these concerns during the rebuttal stage, the authors doubled down by explaining that this choice is motivated by (Sharma et al, 2022). This is not sufficient motivation for why this specific domain should be used, and not a more general set. Similarly, it would not require using a different domain to address some of these concerns -- the distribution of maps could simply be changed such that they are not optimal for GenPlan.

__Concerns about comparisons to alternative models__
* Several reviewers noted concerns about comparisons to alternative models, either as explanations of human planning, or as alternative planning algorithms. While the authors incorporated references into related work, they did not incorporate them into the actual comparisons to human planning. Since some of these are cognitive models (e.g. Ho et al), they could be compared to. The argument that Ho et al deals only with unstructured maps is not a reason to not include it -- DepthPlan is similarly not focused on structure. Similarly, a comparison with Parsel et al (who use an LLM for planning without program synthesis) would be informative for disentangling the role of priors from the role of programs.

__Concerns about generalization (to uncertain domains, and to domains that are sub-optimal for GenPlan)__
* Several reviewers noted issues about generalization, both to domains with uncertainty, and to alternative LLMs. The authors addressed concerns about generalization to other LLMs, but did not directly address questions about uncertainty (beyond noting that it should work theoretically).

Given these concerns, I recommend that the paper is rejected in its current form. Including additional models for comparison, or running the experiment on one further domain, would substantially improve the paper for a future submission.

**Reviewer Concerns:**

__Addressed concerns__
* Robustness of the approach to different LLMs and different prompting strategies
* Incorporation and expansion of related work (for hierarchical modeling, and using LLMs for planning)
* Additional complexity analyses

__Unaddressed concerns__
* Generalization of findings to a second environment (or a different set of mazes that are not explicitly optimal for GenPlan)
* Inclusion of baseline model results (either as cognitive models, or as alternative planning accounts)

**Reviewer Scores:**

Reviewer kTcK
* may have raised score from a 2 to a 4, given that some of their concerns about LLM generalization properties and related work were addressed by rebuttal. However, since authors did not include any other domains in the rebuttal, it seems unlikely that they would vote to accept.

Reviewer mVRx
* may have raised score from a 2 to a 4, but that seems unlikely. The reviewer raised concerns about comparisons to baseline models of hierarchical learning, which the authors did not address, as well as concerns about generalization to other environment settings, which was also unaddressed.

Reviewer sVXi
* unlikely to have changed score (already voted to accept, and authors only partially addressed concerns.)

Reviewer rQQ4
* unlikely to have changed score. Several concerns about presentation of work, baseline model comparisons, and framing, were unaddressed by the rebuttal.

---

### Decision · Program_Chairs · 2026-01-26

Reject